# Population-Aware Imitation Learning in Mean-field Games with Common Noise

Grégoire Lambrecht [1]    Mathieu Laurière [2]

## Abstract

Mean Field Games (MFGs) provide a powerful framework for modeling the collective behavior of large populations of interacting agents. In this paper, we address the problem of Imitation Learning (IL) in MFGs subject to *common noise*, where the population distribution evolves stochastically. This stochasticity compels agents to adopt *population-aware policies* to respond to aggregate shocks. We formulate two distinct learning objectives: recovering a Nash equilibrium and maximizing performance against an expert population. We investigate two imitation proxies: Behavioral Cloning (BC) and Adversarial (ADV) divergence. We then establish finite-sample error bounds showing that minimizing these proxies effectively controls both the policy's exploitability and its performance gap relative to the expert. Furthermore, we propose a numerical framework using generalized Fictitious Play and Deep Learning to compute expert population-aware policies. Through experiments on three environments we demonstrate that standard population-unaware policies fail to capture the equilibrium dynamics. Our results highlight that learning population-aware policies is crucial to avoid being misled by the randomness inherent in common noise.

## 1. Introduction

Mean Field Games (MFGs), introduced independently by (Lasry & Lions, 2007) and (Huang et al., 2006), have emerged as a fundamental framework for analyzing systems with a very large number of rational, indistinguishable

agents. By taking the limit as the number of agents $N \to \infty$, the interactions are simplified into a game between a single representative agent and the aggregate population distribution. While classical MFGs assume that the population distribution evolves deterministically (see e.g. (Bensoussan et al., 2013; Carmona & Delarue, 2018a)), many real-world applications are subject to systemic shocks affecting the entire population simultaneously. Examples include macroeconomics (Lasry et al., 2008; Guéant et al., 2010; Vu & Ichiba, 2025), financial markets (Carmona et al., 2015; 2017; Firoozi et al., 2018; Fu et al., 2021; Burzoni & Campi, 2023; Bassou et al., 2024), price formation (Graber, 2016; Gomes et al., 2023), electricity management (Alasseur et al., 2020; Lavigne & Tankov, 2023; Escribe et al., 2024; Dumitrescu et al., 2024) and crowd motion (Achdou & Lasry, 2018; Perrin et al., 2020). These scenarios are modeled by MFGs with **common noise**, where the mean field becomes a stochastic process adapted to the common noise filtration, see (Carmona et al., 2016; Carmona & Delarue, 2018b) for the theoretical background. Solving such games is significantly more challenging, because the mean field is *stochastic*.

Although the theoretical framework is developed and various applications have been proposed, so far there are no works about learning common noise MFG models from data. This work is a first step in this direction. Here, we focus on the problem of **Imitation Learning (IL)** in MFGs with common noise. We consider a scenario where a learner observes a population of experts playing a Nash equilibrium but does not know the reward function driving their behavior. The goal is to learn a policy that imitates the expert, ideally recovering a Nash equilibrium and matching the expert's performance.

**Related Work:** IL and Inverse Reinforcement Learning (IRL) are well-established in single-agent settings (Ng et al., 2000; Ho & Ermon, 2016). In the context of Multi-Agent Reinforcement Learning (MARL) and MFGs, recent works have begun to address the learnability of equilibria. To the best of our knowledge, Subramanian & Mahajan (2019), Guo et al. (2019), and Elie et al. (2020) were the first to develop and analyze RL for MFGs, aiming at computing Nash equilibria using RL methods and a simulator for the MFG model. RL for MFGs with common noise was studied in (Perrin et al., 2020; Wu et al., 2025). Here, we consider the *converse* problem: learning a policy by looking at data

---

[1] Center for Data Science, New York University and NYU Shanghai, New York, United States of America [2] Shanghai Center for Data Science; NYU-ECNU Institute of Mathematical Sciences at NYU Shanghai; NYU Shanghai, Shanghai, People's Republic of China. Correspondence to: Grégoire Lambrecht <gl3048@nyu.edu>, Mathieu Laurière < mathieu.lauriere@nyu.edu>.

*Proceedings of the 43rd International Conference on Machine Learning*, Seoul, South Korea. PMLR 306, 2026. Copyright 2026 by the author(s).

generated by an expert. Along this line, Ramponi et al. (2023) provide a theoretical foundation for IL in *deterministic* MFGs. We generalize these results to the case with common noise.(Anahtarci et al., 2025; Chen et al., 2022; Zhao et al., 2025) proposed methods to learn policies and (Chen et al., 2024) methods to learn rewards, but so far there are no such works with *common noise*.

**Challenges:** First, common noise breaks the deterministic nature of the mean-field flow, requiring a **probabilistic** analysis of the error. Second, this implies that vanilla policies functions of the individual state only are sub-optimal, requiring agents to adopt **adaptive** policies that react to the fluctuating distribution of the population.

**Contributions:** We provide the first comprehensive **theoretical** and **algorithmic** framework for imitation learning in MFGs with **common noise**. Our main contributions are:

1. **Theoretical Guarantees:** We analyze two imitation metrics: a local Behavioral Cloning (BC) proxy and a global Adversarial (ADV) proxy. We prove that minimizing these proxies guarantees the learned policy is an approximate Nash equilibrium and achieves a reward comparable to the expert.

2. **Algorithmic Framework for Equilibria:** Computing a "ground truth" expert policy in common noise settings is non-trivial. We generalize the Fictitious Play algorithm to the common noise setting. To resolve the issue that Fictitious Play yields a sequence of policies rather than a single deployable one, we introduce a method to distill this sequence into a single population-dependent policy.

3. **Population-aware Imitation:** We highlight the distinction between *vanilla* policies (ignoring the mean field) and *adaptive* policies (observing it). We propose an interactive imitation learning algorithm that trains agents to internalize the relationship between common noise shocks, population shifts, and optimal actions.

The remainder of the paper is organized as follows. Sec. 2 formalizes the framework. Sec. 3 presents our main theoretical bounds linking imitation proxies to equilibrium quality. Sec. 4 details the numerical methods and experimental results. Sec. 5 concludes the paper.

## 2. Problem Setting

### 2.1. Mean Field Dynamics and Nash Equilibria

**Notation.** $\Delta_{\mathcal{S}}$ denotes the set of probability measures over any finite set $\mathcal{S}$. For any integer $N$, $[N] = \{0, \ldots, N\}$ and $[N]^* = [N] \setminus \{0\}$.

**Main components.** A **Mean Field Game (MFG)** with common noise is denoted by a tuple $\mathcal{M} = (\mathcal{X}, \mathcal{A}, \mathcal{E}^0, \nu^0, P, r, H, \rho_0)$, where $\mathcal{X}$ is a finite state space, $\mathcal{A}$ is a finite action space, $\mathcal{E}^0$ is a Polish space for the **common noise** impacting the whole population, $\nu^0$ is a probability measure for the common noise, while $P : \mathcal{X} \times \mathcal{A} \times \Delta_{\mathcal{X}} \times \mathcal{E}^0 \to \Delta_{\mathcal{X}}$ is a stochastic kernel for the agents' dynamics, the mapping $r : \mathcal{X} \times \mathcal{A} \times \Delta_{\mathcal{X}} \to \mathbb{R}$ is a measurable reward function, $H \in \mathbb{N}^*$ is the time horizon, and $\rho_0 \in \Delta_{\mathcal{X}}$ is the initial probability measure. We define a **strategy** as a measurable mapping from $\mathcal{X} \times \Delta_{\mathcal{X}}$ to $\Delta_{\mathcal{A}}$, and a **policy** as a sequence of $H$ strategies. A strategy is denoted by $\pi_0$ and the set of all such strategies is denoted by $\Pi_0$. A policy is denoted by $\pi = (\pi_0, \ldots, \pi_{H-1})$, and the set of all such policies is denoted by $\Pi$. Elements of $\Delta_{\mathcal{X}}$ and $(\Delta_{\mathcal{X}})^H$ are referred to as **mean field** and **mean field sequences** respectively.

**Distribution evolution.** We consider *two types* of distributions: the distribution of the whole population using a given policy, and the distribution of a single agent's state using a given policy when the population is possibly using a different policy. For any pair of population distributions $\rho, \rho' \in \Delta_{\mathcal{X}}$, any strategy $\pi_0 \in \Pi_0$, we define the mean-field transition operator $\Phi(\rho', \pi_0; \rho, e^0)$ as

$$\Phi(\rho', \pi_0; \rho, e^0) = \sum_{x \in \mathcal{X}, a \in \mathcal{A}} \rho'(x)\pi_0(a|x, \rho)P(\cdot|x, a, \rho, e^0).$$

This operator $\Phi$ represents the evolution of the probability distribution of an agent, given that the agent's state distribution was $\rho'$, the agent used strategy $\pi_0$, the total population was distributed according to $\rho$, and the common noise value is $e^0$. We consider an independent and identically distributed (i.i.d.) sequence of random variables $\varepsilon_1^0, \ldots, \varepsilon_{H-1}^0 \in \mathcal{E}^0$ for the common noise. We denote the common noise trajectory up to time $H - 1$ as $\varepsilon^0 = (\varepsilon_1^0, \ldots, \varepsilon_{H-1}^0)$. For any policy $\pi \in \Pi$, we define the mean field sequence $\rho^{\pi, \varepsilon^0} = (\rho_0^{\pi, \varepsilon^0}, \ldots, \rho_{H-1}^{\pi, \varepsilon^0})$ by the recursion: $\rho_0^{\pi, \varepsilon^0} = \rho_0$ and for $n \in [H - 2]$,

$$\rho_{n+1}^{\pi, \varepsilon^0} = \Phi(\rho_n^{\pi, \varepsilon^0}, \pi_n \,;\, \rho_n^{\pi, \varepsilon^0}, \varepsilon_{n+1}^0).$$

This sequence represents the evolution of the population distribution when all agents follow policy $\pi$ and are affected by the common noise trajectory $\varepsilon^0$. For any pair of policies $(\pi, \pi') \in \Pi^2$, we define the mean field sequence $\rho^{\pi, \pi', \varepsilon^0} = (\rho_0^{\pi, \pi', \varepsilon^0}, \ldots, \rho_{H-1}^{\pi, \pi', \varepsilon^0})$ by: $\rho_0^{\pi, \pi', \varepsilon^0} = \rho_0$ and for $n \in [H - 2]$,

$$\rho_{n+1}^{\pi, \pi', \varepsilon^0} = \Phi(\rho_n^{\pi, \pi', \varepsilon^0}, \pi' ; \rho_n^{\pi, \varepsilon^0}, \varepsilon_{n+1}^0).$$

This sequence represents the evolution of the state distribution of a *single agent* following policy $\pi'$ when all other agents follow policy $\pi$ and are affected by the common noise trajectory $\varepsilon^0$.

**Best response and Nash equilibrium.** When the population of agents follows the policy $\pi$, a single agent wishes to find a **best response** to $\pi'$, i.e., a policy maximizing the **value function**:

$$V(\pi', \pi) = \mathbb{E}\left[\sum_{n=0}^{H-1} r(X_n, \alpha_n, \rho_n^{\pi, \varepsilon^0})\right],$$

where the process is defined by $X_0 \sim \rho_0$, $X_{n+1} \sim P(\cdot | X_n, \alpha_n, \rho_n^{\pi, \varepsilon^0}, \varepsilon_{n+1}^0)$, and $\alpha_n \sim \pi'(\cdot | X_n, \rho_n^{\pi, \varepsilon^0})$. Note that under these conditions, the law of the state $X_n$ (conditioned on the common noise) is $\rho_n^{\pi, \pi', \varepsilon^0}$. We denote the law of state-action pair $(X_n, \alpha_n)$ by $\mu_n^{\pi, \pi', \varepsilon^0} \in \Delta_{\mathcal{X} \times \mathcal{A}}$. If $\pi = \pi'$, we will simply denote $\mu_n^{\pi, \varepsilon^0} = \mu_n^{\pi, \pi', \varepsilon^0}$. In the sequel, we will often omit to write explicitly the common noise $\varepsilon^0$, but the sequences $\rho$ and $\mu$ are random variables due to $\varepsilon^0$.

**Definition 2.1.** *For $\pi \in \Pi$, the set of best responses is defined by $BR(\pi) = \arg\max_{\pi' \in \Pi} V(\pi', \pi)$.*

We will use the concept of Nash equilibrium: a situation where no agent has an incentive to unilaterally deviate from their chosen policy.

**Definition 2.2.** *A policy $\pi \in \Pi$ is a* **(mean field) Nash equilibrium** *if $\pi \in BR(\pi)$.*

This is a fixed point-condition: it implies that when the entire population follows policy $\pi$, the resulting mean-field flow $\rho^{\pi}$ creates a reward and transition structure such that the same policy $\pi$ remains optimal for any individual agent.

### 2.2. Imitation in Unknown Reward Environments

Our objective is as follows. Assume that an observer sees realizations of an expert policy $\pi^E$ that constitutes a Nash equilibrium for a MFG $\mathcal{M}$. We further assume that the instantaneous reward function $r$ is unknown to the observer. The observer tries to imitate $\pi^E$ and computes a candidate policy $\pi^A$. We wish to study two questions:

**(Q1)** *How far is $\pi^A$ from being a Nash equilibrium?*

**(Q2)** *How far is $\pi^A$ from being a best response to $\pi^E$?*

The first corresponds to a scenario where a whole population would be using $\pi^A$ while the second corresponds to a scenario where the population is playing $\pi^E$ and a single agent uses policy $\pi^A$. Specifically, we aim to answer these questions in quantitative way using two divergences calculated relative to the expert policy $\pi^E$. The first, $\delta^{BC}$, captures the local discrepancy in decision-making (Behavioral Cloning), while the second, $\delta^{ADV}$, captures the global divergence in the induced state-action distributions (Adversarial).

We define the **Behavioral Cloning (BC) divergence** at time $n$ as:

$$\delta_n^{BC} = \mathbb{E}\left[\mathbb{E}_{X \sim \rho_n^{\pi^E}}\left(\|\pi_n^A(X, \rho_n^{\pi^E}) - \pi_n^E(X, \rho_n^{\pi^E})\|_1\right)\right],$$

where the expectation $\mathbb{E}$ is taken over $\varepsilon_{1:H-1}^0$. We define the **Adversarial (ADV) proxy** at time $n$ as:

$$\delta_n^{ADV} = \mathbb{E}\left[\|\mu_n^{\pi^E} - \mu_n^{\pi^A}\|_1\right].$$

We denote the maximum divergences over the horizon as $\delta^{ADV} = \max_{n \in [H-1]} \delta_n^{ADV}$ and $\delta^{BC} = \max_{n \in [H-1]} \delta_n^{BC}$. We will often use the notation $\rho^E = \rho^{\pi^E}, \mu^E = \mu^{\pi^E}$ and $\rho^A = \rho^{\pi^A}, \mu^A = \mu^{\pi^A}$.

To answer the two questions above, we introduce the following regularity assumptions.

**Assumption 2.3.** *$r$ and $P$ are Lipschitz-continuous w.r.t. the mean-field with constant $L_r, L_P \geq 0$.*

**Assumption 2.4.** *There exists $r_{\max} > 0$ satisfying for any $\pi \in \Pi$:*

$$\mathbb{P}(|r(x, a, \rho_n^{\pi, \varepsilon^0})| \leq r_{\max}) = 1, \quad \forall (x, a) \in \mathcal{X} \times \mathcal{A}.$$

**Definition 2.5** (Mean-Field Lipschitz Policies). *We define the class of* **Mean-Field Lipschitz Policies**, *denoted by $\Pi_{lip}^{\mathcal{M}}$, as the set of policies $\pi$ for which there exists a constant $L_E \geq 0$ such that for any sequence of policies $(\pi', \pi'', \pi^1, \pi^2, \pi^3, \pi^4) \in \Pi^6$ and any time step $n \in [H-1]$, the following inequality holds:*

$$\mathbb{E}\left[\mathbb{E}_{X \sim \rho_n^{\pi', \pi''}}\left(\|\pi(x, \rho_n^{\pi^1, \pi^2}) - \pi(x, \rho_n^{\pi^3, \pi^4})\|_1\right)\right]$$
$$\leq L_E \mathbb{E}\left[\|\rho_n^{\pi^1, \pi^2} - \rho_n^{\pi^3, \pi^4}\|_1\right].$$

We will assume that our policy space $\Pi$ is restricted to a compact subset of $\Pi_{lip}^{\mathcal{M}}$. This is a well-founded assumption, as the set of policies that are Lipschitz-continuous with a fixed constant $L_E$ with respect to the mean-field is indeed compact. From this point forward, we will write $\Pi$ to refer to this compact set of Lipschitz policies, utilizing an abuse of notation for the sake of brevity.

**Definition 2.6.** *The* **exploitability** *of $\pi \in \Pi$ is defined as:* $\mathcal{E}(\pi) = \max_{\pi' \in \Pi} V(\pi', \pi) - V(\pi, \pi)$.

Exploitability serves as a direct measure of the proximity to be an equilibrium: $\pi$ is a Nash equilibrium iff $\mathcal{E}(\pi) = 0$, while a value close to zero indicates that the policy is nearly optimal for each agent.

**Definition 2.7.** *For $\epsilon \geq 0$, a policy $\pi \in \Pi$ is an $\epsilon$-**Nash equilibrium** if $\mathcal{E}(\pi) \leq \epsilon$.*

In the context of recovering an equilibrium from the policy $\pi^E$, we want to control $\mathcal{E}(\pi)$ with $\delta^{BC}$ and $\delta^{ADV}$.

## 3. Imitation Learning Results

We first address **(Q1)**. Due to space constraints, we provide the results and the main ideas of the proofs in Appendix A.

**Theorem 3.1.** *Suppose that Assm 2.3 and 2.4 hold. We have*

$$\mathcal{E}(\pi^A) \leq \delta^{BC}\Big[r_{\max}H^2$$
$$+ 2\frac{(1 + L_P + L_E)^H}{(L_P + L_E)^2}(L_r + r_{\max}(L_E + 1))\Big].$$

**Theorem 3.2.** *Suppose Assm 2.3 and 2.4 hold. Then,*

$$\mathcal{E}(\pi^A) \leq \delta^{ADV}\Big[H(2L_r + 3L_E r_{\max} + r_{\max})$$
$$+ 3r_{\max}H^2(L_E + L_P)\Big].$$

**Remark 3.3.** *Our results are consistent with the ones established in (Ramponi et al., 2023). By letting $L_E = 0$ in Theorem 3.1, one obtains (Ramponi et al., 2023, Theorem 3). We recover (Ramponi et al., 2023, Theorem 5) when $L_E = 0$ in Theorem 3.2.*

**Remark 3.4.** *We do not address the case where $L_P = 0$ and $L_E = 0$. In this specific setting, the dynamics and the expert policy are independent of the population distribution. Under these conditions, it is straightforward to observe that the results from Ramponi et al. (2023, Theorem 1) and Ramponi et al. (2023, Theorem 2) directly apply.*

We then turn to **(Q2)** and give the guarantees of the performance of the policy $\pi^A$ versus $\pi^E$.

**Theorem 3.5.** *Suppose that Assm 2.4 holds. We have*

$$|V(\pi^E, \pi^E) - V(\pi^A, \pi^E)| \leq H^2\delta^{BC}r_{\max}.$$

**Theorem 3.6.** *Suppose Assm 2.3 and 2.4 hold. We have*

$$|V(\pi^E, \pi^E) - V(\pi^A, \pi^E)| \leq r_{\max}\delta^{ADV}\Big[H(L_E + 1)$$
$$+ H^2(L_P + L_E)\Big].$$

**Remark 3.7.** *In Theorem 3.6, Assumption 2.3 is required only for the Lipschitz continuity of the transition kernel $P$, but not for the reward $r$.*

**Results Interpretation and Analysis:** Our results demonstrate that answering Question **(Q1)** (Equilibrium Recovery) is significantly more subtle than answering Question **(Q2)** (Performance Matching).

Robustness of Performance **(Q2)**: As shown in Theorem 3.5, the performance gap is bounded by $\delta^{BC}$ without requiring Lipschitz continuity for the reward ($L_r$) or transition kernel ($L_P$). Even in complex or discontinuous environments, local action mimicry is sufficient to achieve expert-level utility.

Fragility of Equilibrium **(Q1)**: This stability does not hold for equilibrium recovery. An agent may achieve high rewards while remaining far from a Nash equilibrium. Theorem 3.1 shows that exploitability $\mathcal{E}(\pi^A)$ is sensitive to social reactivity ($L_E$) and environmental coupling ($L_P$), resulting in exponential error growth.

Finally, our results suggest that the ADV divergence is more robust for solving **(Q1)** (Equilibrium Recovery) than BC divergence, because it transforms the error propagation from exponential (Theorem 3.1) to polynomial (Theorem 3.2).

## 4. Numerical Experiments

### 4.1. Objectives and Method

We aim to illustrate the control of the Behavioral Cloning (BC) and Adversarial (ADV) proxies on the performance relative to the expert and the equilibrium imitation learning problem. To this end, we first compute a Nash equilibrium, designated as the expert policy $\pi^E$. The objective of the learner is twofold: to obtain a high reward by matching the expert's performance and to learn a Nash equilibrium policy. We compare the case where the learner explicitly accounts for common noise or not. Thus, we derive two policies from agent trajectories generated by this expert: **(1)** a population-*un*aware **vanilla policy** $\hat{\pi}^{\text{vanilla}}$, and **(2)** a population-*aware* **adaptive policy** $\hat{\pi}^{\text{adaptive}}$; so $\hat{\pi}_n^{\text{vanilla}}$ : $\mathcal{X} \to \Delta_{\mathcal{A}}$ while $\hat{\pi}_n^{\text{adaptive}}$ : $\mathcal{X} \times \Delta_{\mathcal{X}} \to \Delta_{\mathcal{A}}$. We propose two methods to obtain these policies. First, a method well-suited for small spaces and based on Nadaraya-Watson kernel regression. Second, an interactive imitation approach, where a single learning agent is placed within an environment populated by other agents following the expert policy. In all the examples, the proxies will be computed via a Monte Carlo procedure described in Algorithm 4 in Appx. B.

### 4.2. Challenges

**Nash Equilibrium:** Computing Nash equilibrium for MFGs without common noise is a well-understood problem in the literature (Achdou & Laurière, 2020). However, a gap remains concerning MFGs with common noise: Most existing works consider common noise path-dependent policies, e.g. (Achdou & Lasry, 2018; Perrin et al., 2022). However, this becomes intractable as the dimension of the common noise and the time horizon increase.

**Imitation Learning:** (Chen et al., 2022; Anahtarci et al., 2025; Zhao et al., 2025) proposed methods to learn policies for MFGs without common noise, but there is no such work with common noise. Here, we need to deal with stochastic mean fields and population-dependent policies.

## 4.3. Environment and Metrics

We consider three different examples with different effects of common noise. The first example is a simple two-state, two-action space. The second example is the Beach Bar environment, and the last example is the Night Club environment inspired by a social choice model (Salhab et al., 2019).

The motivation for testing on three different environments is to demonstrate that our approach is efficient in a variety of structures of common noise:

- **Exp. 1, Concentration-Dependent:** The common noise is correlated with the population density (e.g., if the population is concentrated at a state, agents are more likely to be affected).

- **Exp. 2, Local Effects:** The noise affects agent positions heterogeneously, leading to more chaotic local shifts.

- **Exp. 3, Block-Shift:** The common noise shifts specific clusters or "blocks" of agents while leaving others unaffected, representing a segmented environmental shock.

In real-world applications, common noise is complex and difficult to model explicitly. By showing consistent results across these three distinct scenarios, we validate that our population-aware imitation learning method remains superior regardless of the specific common noise manifestation.

After computing an expert policy $\pi^E$, a candidate policy $\pi^A$ (which can be either the vanilla policy $\hat{\pi}^{\text{vanilla}}$ or the adaptive policy $\hat{\pi}^{\text{adaptive}}$), we report six metrics:

- **Imitation Proxies:** Behavioral Cloning (BC) and Adversarial (ADV) errors.

- **Performance vs. Expert:** Reward vs. Expert, and Relative Reward vs. Expert: $\mathcal{V}(\pi^A, \pi^E) = [V(\pi^A, \pi^E) - V(\pi^E, \pi^E)]/|V(\pi^E, \pi^E)|$.

- **Equilibrium Quality:** Exploitability, and Relative Exploitability: $\mathcal{E}(\pi^A, \pi^E) = [\mathcal{E}(\pi^A)]/|V(\pi^E, \pi^E)|$.

We normalize reward and exploitability metrics to obtain quantities that can be compared across environments.

To ensure statistical stability, for each parameter configuration, we perform 5 independent runs of the full experimental pipeline: computing the expert equilibrium, estimating the vanilla and adaptive policies, and calculating the performance metrics.

---

**Algorithm 1** Fictitious Play for MFG with Common Noise

**Require:** MFG $\mathcal{M} = (\mathcal{X}, \mathcal{A}, \mathcal{E}^0, \nu^0, P, r, H, \rho_0)$; Initial policy $\pi^0 \in \Pi$; Number of iterations $K$.
1: **for** $k = 1, \ldots, K$ **do**
2:    Train $\pi^k$ as the best response to $(\pi^0, \ldots, \pi^{k-1})$.
3: **end for**
4: **Return** policy tuple $(\pi^0, \ldots, \pi^K)$.

---

### 4.4. Fictitious Play for MFGs with Common Noise

As noted earlier, the literature on algorithms for computing Nash equilibria with population-dependent policies for MFGs with common noise is rather scarce. Here we adapt to situations with common noise the Fictitious Play algorithm proposed in (Perrin et al., 2022) to obtain a population-dependent Nash equilibrium policy referred to as a **Master policy** (without common noise). At each iteration, a best response is trained against an aggregated population distribution generated by the sequence of policies obtained in previous iterations.

Specifically, given the policies $(\pi^0, \ldots, \pi^k) \in \Pi^{k+1}$ up to iteration $k$, a policy $\pi^{k+1}$ is defined as a best response to the sequence $(\pi^0, \ldots, \pi^k)$ if it maximizes:

$$\pi \mapsto \mathbb{E}\Big[ \sum_{n=0}^{H-1} r(X_n, \alpha_n, \bar{\rho}_n^{k,\varepsilon^0}) \Big],$$

where $(X_n, \alpha_n)$ is induced by $\pi$ and the **aggregate mean-field** sequence $\bar{\rho}^k$ is defined as

$$\bar{\rho}_n^{k,\varepsilon^0} = \frac{1}{k+1} \sum_{i=0}^{k} \rho_n^{\pi^i,\varepsilon^0}. \tag{1}$$

The generalized Fictitious Play procedure is summarized in Alg. 1.

To obtain the best response at each iteration, we parametrize the policy by a neural network (NN), and train the NN using Alg. 5 by directly maximizing the value function, see Appx. B. Note that Alg. 1 returns a list of policies instead of a single, equilibrium policy. To compute $\bar{\rho}^K$, one can average results obtained with each policy $\pi^k$ of the list, but this is not sufficient for our purpose since we want to have in hand an expert policy $\pi^E$. To address this, we propose an IL algorithm that trains a NN to reproduce the aggregated mean-field trajectories, as detailed in Alg. 2.

### 4.5. Learning Algorithms

To demonstrate the relevance of BC and ADV divergences in distinguishing between population-aware and population-unaware policies, we propose a framework capable of learning both policy types.

Since our primary objective is to illustrate the efficacy of BC and ADV metrics, we introduce a simple IL algorithm, see

**Algorithm 2** Mean Field Imitation Learning (MF-IL)

**Require:** MFG $\mathcal{M}$; Policy sequence $(\pi^0, \ldots, \pi^K)$; Initial parameters $\theta_0$ for parametric policy $\pi_\theta$; Iteration number $J$; Batch size $B$.

1: **for** $j = 1, \ldots, J-1$ **do**
2:    Sample common noises $(\varepsilon^{0,s})_{1 \leq s \leq B}$.
3:    **for** $s = 1, \ldots, B$ **do**
4:       Compute $\bar{\rho}^{K, \varepsilon^{0,s}}$ from $(\pi^0, \ldots, \pi^K)$, see (1)
5:       Generate $\rho^{\pi_{\theta_{j-1}}, \varepsilon^{0,s}}$ induced by $\pi_{\theta_{j-1}}$.
6:    **end for**
7:    Compute $L(\theta_{j-1}) = \frac{1}{B} \sum_{s,t} \|\rho_t^{\pi_{\theta_{j-1}}, \varepsilon^{0,s}} - \bar{\rho}_t^{K, \varepsilon^{0,s}}\|_1$.
8:    Obtain $\theta_j$ by one step of gradient of $L(\theta_{j-1})$.
9: **end for**
10: **Return** Trained parametric policy $\pi_{\theta_J}$.

**Algorithm 3** Interactive Imitation Learning for Population-aware Policies

**Require:** MFG $\mathcal{M} = (\mathcal{X}, \mathcal{A}, \mathcal{E}^0, \nu^0, P, r, H, \rho_0)$; Expert policy $\pi^E$; Initial parameters $\theta_0$; Iteration number $J$; Batch size $B$; Population size $M$.

1: **for** $j = 1, \ldots, J$ **do**
2:    Sample common noise $(\varepsilon_t^{0,s})_{0 \leq t \leq H-1}^{1 \leq s \leq B}$.
3:    **for** $s = 1, \ldots, B$ **do**
4:       Generate $M$ state-action trajectories $\{(x_t^{s,i}, a_t^{s,i})\}_{i=1}^M$ using $\pi^E$ under noise $\varepsilon_{1:H-1}^{0,s}$.
5:       Set empirical MF: $\hat{\rho}_t^{s,M}(x) = \frac{1}{M} \sum_{i=1}^M \delta_{\{x = x_t^{s,i}\}}$.
6:       Set empirical strategy: $\hat{\pi}_t^{s,M}(a|x) = \frac{1}{M \hat{\rho}_t^{s,M}(x)} \sum_{i=1}^M \delta_{\{(x,a) = (x_t^{s,i}, a_t^{s,i})\}}$.
7:    **end for**
8:    Define loss: $L(\theta_{j-1}) = \frac{1}{BM} \sum_{s,t,i} \|\pi_t^{\theta_{j-1}}(x_t^{s,i}, \hat{\rho}_t^{s,M}) - \hat{\pi}_t^{s,M}(x_t^{s,i})\|_1$.
9:    Obtain $\theta_j$ by one step of gradient of $L(\theta_{j-1})$.
10: **end for**
11: **Return** Trained adaptive policy $\pi_{\theta_J}$.

Alg. 3. This setup mirrors a scenario where a single agent interacts with an environment populated by expert agents who are already following a Nash equilibrium policy $\pi^E$. We assume the learner can observe the states and actions of these expert agents, allowing it to internalize the relationship between the common-noise-induced mean-field $\rho$ and the resulting expert actions. For population-unaware policies, the same algorithm can be used but removing the mean-field input in $\pi^{\theta_j}$.

### 4.6. First Experiment: 2 States, 2 Actions Environment

**Model:** Let $\mathcal{X} = \{0, 1\}$, $\mathcal{A} = \{0, 1\}$, $\mathcal{E}^0 = [0, 1]$, and initial distribution $\rho_0 = (\frac{1}{2}, \frac{1}{2})$. The dynamics are as follows: For any $e^0 \in [0, 1]$ and $\rho \in \Delta_{\mathcal{X}}$, we introduce the *perturbed* probability measure $[e^0 \rho]$ defined by: $[e^0 \rho](1) = \frac{e^0 \rho(1)}{e^0 \rho(1) + (1-e^0)\rho(0)}$. For any $\eta \in [0, 1]$ and for all

$(x, a, \rho, e^0) \in \mathcal{X} \times \mathcal{A} \times \Delta_{\mathcal{X}} \times [0, 1]$, $P_\eta(\cdot|x, a, \rho, e^0) = (1 - \eta)\delta_a + \eta[e^0 \rho]$. The reward is: $r(x, a, \rho) = -\mathbb{1}_{\{x=0\}}\rho(0) - \mathbb{1}_{\{x=1\}}\rho(1)$. For the common noise distribution, we consider $\nu_\alpha^0 = \text{Beta}(\alpha, \alpha)$. This model represents agents who choose between two alternatives under congestion effects. Individual decisions are occasionally ignored and replaced by a population-level force. When this force is active, its impact is governed by $\alpha$. When $\alpha < 1$, $\text{Beta}(\alpha, \alpha)$ is U-shaped and $[\varepsilon_1^0 \rho]$ concentrates at 0 or 1, with equal probability. When $\alpha = 1$, $\text{Beta}(1, 1)$ is the uniform law, and the behavior of $[\varepsilon_1^0 \rho]$ is erratic. When $\alpha > 1$, $\text{Beta}(\alpha, \alpha)$ is concentrated around $\frac{1}{2}$ and the perturbation $[\varepsilon_1^0 \rho]$ stays close to $\rho$. In short, $\eta$ controls the frequency at which the common noise impacts the agents, and $\alpha$ controls its effect.

**Expert Policy:** To compute a Nash equilibrium on this simple environment, we use an ad-hoc method inspired by Mann iterations of best responses. The best response is computed via dynamic programming following (Gu et al., 2024), adapted to incorporate common noise. This method is made possible by the simplicity of the environment; details of the algorithm are presented in Appx. C. The relevance of this method is justified by the low exploitability achieved by the expert policy. For each tuple of parameters $(\alpha, \eta) \in \mathbb{R}_+ \times [0, 1]$, we denote the expert policy as $\pi_{\alpha, \eta}^E$.

**Vanilla and Adaptive Policies:** The simplicity of the environment allows us to consider classical statistical estimators for the vanilla and adaptive policies. For this example, we propose a method based on Nadaraya-Watson kernel regression. To this end, we sample synthetic state-action trajectories of agents following the expert policy under the same common noise realizations. This allows us to regress the expert policy by observing the actions alongside the empirical population distribution. The adaptive policy thus interpolates the expert's behavior across the continuous space of mean-field realizations, while the vanilla policy serves as a baseline that ignores common noise fluctuations. Complete technical details are provided in Appx. C.

**Results:** We consider $\alpha \in \{0.75, 1, 1.25, 1.5, 1.75\}$ and $\eta \in \{0, 0.25, 0.5, 0.75, 1\}$.

The complete grid of results is provided in Appx. C. We highlight specific findings for $\eta = 0.75$ in Figure 1 and for $\alpha = 1.75$ in Figure 11b in Appx. C. The results indicate that the problem exhibits significant parameter sensitivity. Notably, the adaptive policy consistently outperforms the vanilla policy across all tested regimes.

This performance gap can be explained by a qualitative analysis of the learned policies. Figure 2 illustrates the behavior of $\pi^E$, $\hat{\pi}^{\text{vanilla}}$, and $\hat{\pi}^{\text{adaptive}}$ as a function of the mean-field variable $\rho$. While the expert dynamically adapts its actions to the fluctuations of the mean-field, the adaptive policy successfully reconstructs this dependency. In contrast, the

vanilla policy only captures the average action of the expert.

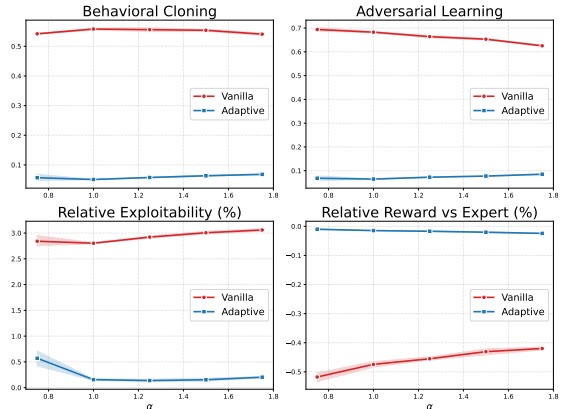

*Figure 1.* Performance metrics for 5 different runs, with $\eta = 0.75$.

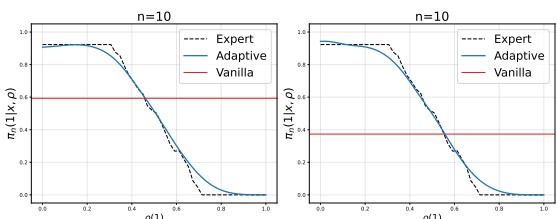

*Figure 2.* Policy comparison for $\alpha = 1.75$, $\eta = 0.75$.

### 4.7. Second Experiment: The Beach Bar Environment

We consider the *Beach Bar* environment as presented in (Perrin et al., 2022). In this setting, the bar is permanently open and the agents are subject to a common noise.

**Model:** Let $\mathcal{X} = \{0, \ldots, 4X\}$, $\mathcal{A} = \{-1, 0, 1\}$, and $\mathcal{E}^0 = \mathcal{X}^{|\mathcal{X}|}$. For parameters $\alpha, \beta, \eta$, and common noise $e^0 \in \mathcal{E}^0$, the dynamics are governed by a transition kernel $P$ that incorporates both *common noise* $e^0(x)$ and *idiosyncratic noise* (represented by a uniform shift $s \in \{-1, 0, 1\}$). Specifically, $P(x, a, \rho, e^0) = \frac{1}{3} \sum_{s \in \{-1,0,1\}} \delta_{x+a+e^0(x)+s \pmod{|\mathcal{X}|}}$. The common noise law is defined as $\varepsilon_1^0 \sim (\nu_\eta^0)^{\otimes |\mathcal{X}|}$, where $\nu_\eta^0$ is the distribution of $\varepsilon^{0,1}\varepsilon^{0,2}$ with $\varepsilon^{0,1} \sim \text{Bernoulli}(\eta)$ and $\varepsilon^{0,2} \sim \text{Unif}\{-X, \ldots, X\}$. The reward function, reflecting a preference for a central bar at $x_{\text{bar}} = 2X$ and a penalty for overcrowding, is: $r(x, a, \rho) = -|x - x_{\text{bar}}| \mathbb{1}_{\{\rho(x_{\text{bar}}) \leq \beta\}} - \alpha \log(\rho(x)) - |a|$. This model represents a beach environment where holidaymakers balance two competing objectives: avoiding crowds while remaining close to the bar. However, the agents are subject to a threshold effect; when the bar becomes too crowded ($\rho(x_{\text{bar}}) > \beta$), it loses its appeal, and agents lose all interest to it. The environment is disrupted by external events that simultaneously displace all agents at specific locations. These may represent physical changes, such as shifting shadows making certain spots undesirable, or social phenomena like spontaneous group movements. These stochastic events are governed by the common noise, where the parameter $\eta$ controls the frequency of such perturbations.

For the following, we fix $\beta = 0.1$. We study the effect of the parameters $(\alpha, \eta)$ on our learning problems.

**Expert Policy:** For this large scale environment, we use our new algorithmic procedure described in Subsec. 4.4. Figure 3 shows one realization of the mean-field trajectory induced by the expert, for $(\alpha, \eta) = (1, 0.3)$.

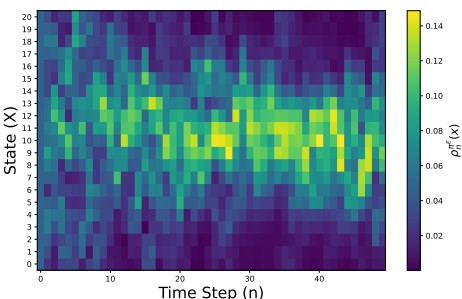

*Figure 3.* One realization of the mean-field trajectory generated by the expert, for $(\alpha, \eta) = (1, 0.3)$.

**Vanilla and Adaptive Policies:** For each value of $(\alpha, \eta)$, we consider learning algorithms presented in Subsec. 4.5. We report all the learning parameters used in Appx. D.

**Convergence of the Algorithms:** Figure 4 reports, for $(\alpha, \eta) = (1, 0.3)$, the convergence of the FP procedure, the expert loss from Algorithm 2, and the losses of the adaptive and vanilla policies from Algorithm 3. The FP procedure is confirmed to converge toward a Nash equilibrium as the number of iterations grows. Moreover, the adaptive policy achieves lower loss than the vanilla policy, indicating superior imitation of the expert. Additional results for $\eta = 0.3$ across multiple values of $\alpha$ are provided in Appx. D, Figure 17.

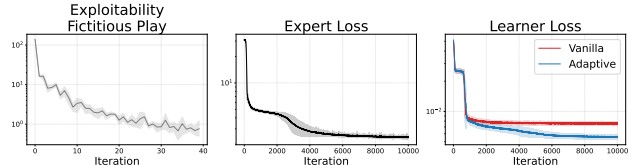

*Figure 4.* Convergence curves for $(\alpha, \eta) = (1, 0.3)$: FP exploitability (left), expert loss (center), and adaptive vs. vanilla losses (right).

**Results:** We take $\eta \in \{0.1, 0.2, 0.3, 0.4, 0.5\}$ and $\alpha \in \{0.1, 0.5, 1, 1.5, 2\}$, $X = 5$ ($|\mathcal{X}| = 20$) and $H = 50$.

The complete grid of results is provided in Appx. D. We highlight specific findings for $\eta = 0.3$ in Figure 5 and $\alpha = 1$ in Figure 14b in Appx. D. The results indicate that the problem exhibits significant parameter sensitivity. We again observe that the adaptive policy consistently outperforms the vanilla policy across all tested regimes.

The qualitative analysis for this environment mirrors that of Example 1. The adaptive policy successfully reconstructs

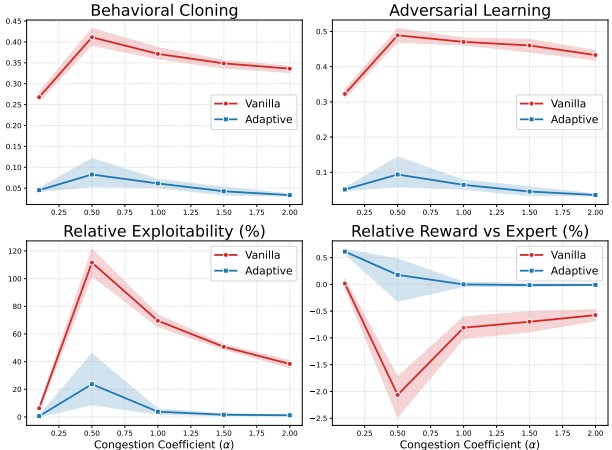

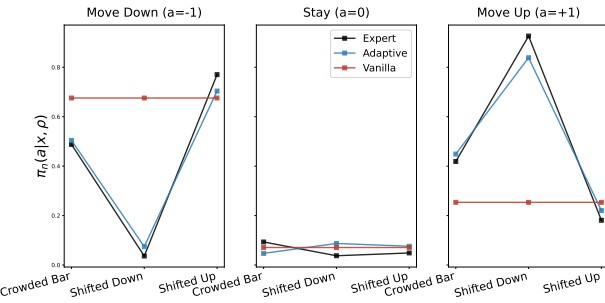

Figure 5. Performance metrics for 5 different runs, with $\eta = 0.3$.

Figure 6. Policy sensitivity analysis for the Beach Bar environment with parameters $(\alpha, \eta) = (1, 0.3)$ at time $n = 10$ and state $x = x_{\text{bar}}$. The comparison illustrates how the Expert (Nash), Adaptive, and Vanilla policies respond to different population distributions: *Crowded Bar* represents high density at the bar location, testing the agent's avoidance behavior; *Shifted Down* and *Shifted Up* reflect scenarios where the mass is concentrated immediately below or above the bar, respectively. This demonstrates the Adaptive policy's ability to react to mean-field fluctuations, whereas the Vanilla policy remains static due to its lack of population awareness.

the expert's dependency on $\rho$, whereas the vanilla policy fails to react to mean-field fluctuations, as we can see in Figure 6. This explains the observed performance disparity.

### 4.8. Third Experiment: The Night Clubs Environment

The following model is inspired by the mean-field linear-quadratic-Gaussian (LQG) model for social choice developed by Salhab et al. (2019). This environment shares structural similarities with the *Beach Bar* problem. However, the reward structure is different: the environment features two distinct points of interest (the "night clubs"), and agents are motivated by a gathering incentive, a desire to congregate rather than avoid congestion. Furthermore, the common noise in this setting is designed to affect agents in a more highly correlated manner than in previous experiments, simulating global trends that shift the popularity of one club over another.

**Model:** Let $\mathcal{X}, \mathcal{A}, \mathcal{E}^0$, and dynamics $P$ be as defined in the Beach Bar model, with clubs at $x_{\text{club}}^1 = X$ and $x_{\text{club}}^2 = 3X$. The reward function $r(x, a, \rho) = -D_\beta^k(x, \rho) - \alpha|x - \bar{x}(\rho)|$ prioritizes a target club based on crowding and penalized distance from the circular mean $\bar{x}(\rho)$. The effective distance is $D_\beta^k(x, \rho) = |x - x_{\text{club}}^k|$ if $\rho(x_{\text{club}}^k) \leq \beta$, and $|x - x_{\text{club}}^{-k}|$ otherwise. Common noise $\varepsilon_1^0$ follows a localized **block-shift** law: at each step, a shift magnitude $s \sim \text{Unif}\{-\lfloor\frac{|\mathcal{X}|}{3}\rfloor, \ldots, \lfloor\frac{|\mathcal{X}|}{3}\rfloor\}$ and a block start $i \sim \text{Unif}(\mathcal{X})$ are sampled. The resulting noise vector is $\varepsilon_1^0(x) = s \cdot \mathbb{1}_{\{x \in \{i, i+1 \pmod{|\mathcal{X}|}\}\}} \cdot \eta^{1/3}$, effectively shifting only a local block of agents. This model represents a social environment where clubgoers balance two competing objectives: the desire for social gathering and the pursuit of entertainment at local venues. Agents seek to minimize their distance to the nearest club to enjoy the music; however, the clubs are subject to a mechanical threshold effect. If a club's local density exceeds the capacity $\beta$ ($\rho(x_{\text{club}}) > \beta$), the venue is considered overcrowded, the music ceases, and the agents lose all incentive to remain at that location. Agents immediately redirect their interest toward the alternative club. This redirection is 'blind', the agents are assumed to be too distant to perceive the current status (open or closed) of the far venue, and thus target its location regardless of its density. The environment is disrupted by structured external events that simultaneously displace groups of agents, modeling sudden interest in the direction of one of the music venues. For the following, we fix $\beta = 0.1$. We study the effect of the parameters $(\alpha, \eta)$ on our learning problems.

**Expert, Vanilla and Adaptive Policies:** We consider the exact same procedure as in the Beach Bar example. Hyperparameter details and training loss are given in Appx. E. Figure 7 shows one realization of the mean-field trajectory induced by the expert, for $(\alpha, \eta) = (0.1, 1)$. Figure 8 reports the convergence of the FP procedure and the imitation losses for the same parameter configuration. As in the Beach Bar experiment, the FP procedure converges toward a Nash equilibrium, and the adaptive policy achieves lower loss than the vanilla policy, confirming its superior imitation of the expert.

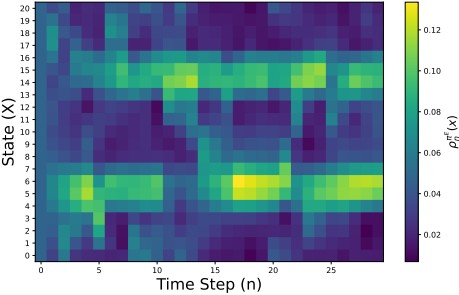

Figure 7. One realization of the mean-field trajectory generated by the expert for $(\alpha, \eta) = (0.1, 1)$.

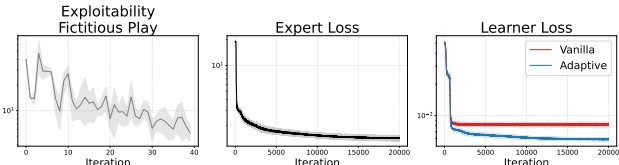

*Figure 8.* Convergence curves for $(\alpha, \eta) = (0.1, 1)$: FP exploitability (left), expert loss (center), and adaptive vs. vanilla losses (right).

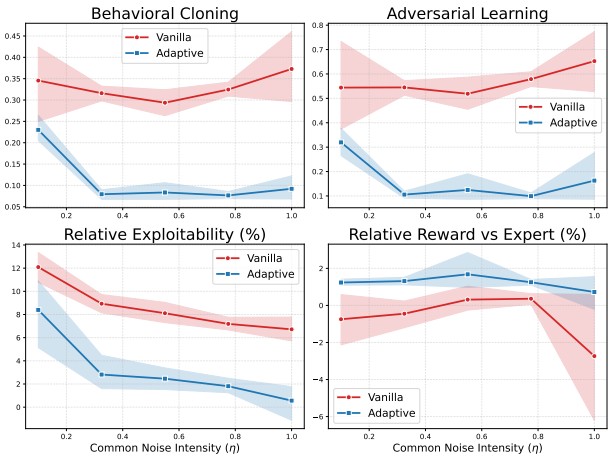

*Figure 9.* Performance metrics for 5 different runs, with $\alpha = 1.05$.

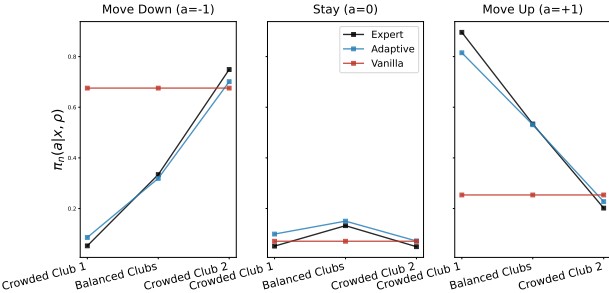

*Figure 10.* Policy sensitivity analysis for the Night Clubs environment with parameters $(\alpha, \eta) = (0.1, 1)$ at time $n = 10$ and state $x = 2X$. The comparison illustrates how the Expert (Nash), Adaptive, and Vanilla policies respond to different population distributions: *Crowded Club 1* represents high density at the Club 1 location; *Balanced Clubs* represents equal density at Club 1 and Club 2 locations. *Crowded Club 2* represents high density at the Club 2 location.

**Results:** We take $\eta \in \{0.1, 0.325, 0.55, 0.775, 1\}$ and $\alpha \in \{0.1, 0.575, 1.05, 1.525, 2\}$, $X = 5$ ($|\mathcal{X}| = 20$) and $H = 30$.

The complete grid of results is provided in Appx. E. We highlight specific findings for $\alpha = 1.05$ in Figure 9 and for $\alpha = 0.1$ in Figure 18b in Appx. E. The results indicate that the problem exhibits significant parameter sensitivity. We again observe that the adaptive policy consistently outperforms the vanilla policy across all tested regimes, but that the advantage on **(Q2)** (reward matching) is less obvious.

The qualitative behavior in the Night Clubs environment

mirrors our previous observations. As shown in Figure 10, the adaptive policy effectively reconstructs the expert's reactivity to mean-field fluctuations $\rho$, whereas the vanilla policy fails to capture these dynamics, serving as a mere average of expert actions.

## 5. Conclusion

We present the first framework for Imitation Learning in Mean Field Games with common noise. By extending Behavioral Cloning and Adversarial proxies to stochastic mean fields, we prove these metrics control both exploitability and performance relative to an expert. Our analysis highlights the critical role of **adaptive policies**, showing that population-unaware strategies fail to capture equilibrium dynamics under stochastic fluctuations. To validate this, we developed a novel pipeline combining generalized Fictitious Play and Deep Learning to approximate population-aware policies. Future work should focus on extending the framework for instance to state-action mean field interactions and applying the method to real-world data.

## Impact Statement

While our work is primarily theoretical, we believe it could find implications in real-world applications involving massive agent interactions. Due to high dimensionality and the difficulty of characterization, scientific models often omit common noise: the external stochastic signals that affect an entire population simultaneously. However, real-world equilibria are influenced by these global signals. In financial markets, for example, a single statement from a political figure on social media can trigger instantaneous, population-wide shifts in trading volumes and asset prices. Or in energy infrastructure, environmental randomness in renewable energy production affects all producers in a geographic region simultaneously. Our results suggest that in those situations, Imitation Learning with population-aware policies is more robust than population-unaware policies.

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

# A. Useful Inequalities and Proofs

In this appendix, we first present the inequalities that are used to establish our theoretical results. We then proceed to the proofs of the theorems.

## A.1. Useful Inequalities

First we extend (Ramponi et al., 2023, Lemma 2 C.4).

**Lemma A.1.** *Under Assumptions 2.3 and 2.4, for any policies $\pi^1, \pi^2, \pi^3 \in \Pi$, we have:*

$$|V(\pi^3, \pi^1) - V(\pi^3, \pi^2)| \leq L_r \mathbb{E}\left[\sum_{n=0}^{H-1} \|\rho_n^1 - \rho_n^2\|_1\right] + r_{\max}\mathbb{E}\left[\sum_{n=0}^{H-1} \|\rho_n^{1,3} - \rho_n^{2,3}\|_1\right]$$
$$+ r_{\max} \sum_{n=0}^{H-1} \mathbb{E}\left[\mathbb{E}_{X \sim \rho_n^{2,3}}\left[\|\pi_n^3(X, \rho_n^1) - \pi_n^3(X, \rho_n^2)\|_1\right]\right].$$

*Proof.* Expanding the value function difference:

$$|V(\pi^3, \pi^1) - V(\pi^3, \pi^2)| = \left|\mathbb{E}\left[\sum_{n=0}^{H-1} \sum_{x,a} \left(\mu_n^{1,3}(x,a)r(x,a,\rho_n^1) - \mu_n^{2,3}(x,a)r(x,a,\rho_n^2)\right)\right]\right|$$
$$\leq \underbrace{\mathbb{E}\left[\sum_{n,x,a} \mu_n^{1,3}(x,a)|r(x,a,\rho_n^1) - r(x,a,\rho_n^2)|\right]}_{A}$$
$$+ \underbrace{\mathbb{E}\left[\sum_{n,x,a} |(\mu_n^{1,3}(x,a) - \mu_n^{2,3}(x,a))r(x,a,\rho_n^2)|\right]}_{B}.$$

For term $A$, by the Lipschitz continuity of $r$ in the mean-field:

$$A \leq \mathbb{E}\left[\sum_{n=0}^{H-1} L_r \|\rho_n^1 - \rho_n^2\|_1 \sum_{x,a} \mu_n^{1,3}(x,a)\right] = L_r \mathbb{E}\left[\sum_{n=0}^{H-1} \|\rho_n^1 - \rho_n^2\|_1\right].$$

For term $B$, we use the definition $\mu_n(x,a) = \rho_n(x)\pi_n(a|x,\rho)$ and the triangle inequality:

$$B \leq r_{\max}\mathbb{E}\left[\sum_{n,x,a} |\rho_n^{1,3}(x)\pi_n^3(a|x,\rho_n^1) - \rho_n^{2,3}(x)\pi_n^3(a|x,\rho_n^2)|\right]$$
$$\leq r_{\max}\mathbb{E}\left[\sum_{n,x,a} \pi_n^3(a|x,\rho_n^1)|\rho_n^{1,3}(x) - \rho_n^{2,3}(x)| + \rho_n^{2,3}(x)|\pi_n^3(a|x,\rho_n^1) - \pi_n^3(a|x,\rho_n^2)|\right]$$
$$= r_{\max}\mathbb{E}\left[\sum_{n=0}^{H-1} \|\rho_n^{1,3} - \rho_n^{2,3}\|_1\right] + r_{\max} \sum_{n=0}^{H-1} \mathbb{E}\left[\mathbb{E}_{X \sim \rho_n^{2,3}}\left[\|\pi_n^3(X, \rho_n^1) - \pi_n^3(X, \rho_n^2)\|_1\right]\right].$$

This concludes the proof. □

Now we verify (Ramponi et al., 2023, Lemma 3 C.4).

**Lemma A.2.** *Recall that $\delta^{BC} = \max_{n \in [H-1]} \mathbb{E}(\mathbb{E}_{X \sim \rho_n^E}(\|\pi_n^E(X, \rho_n^E) - \pi_n^A(X, \rho_n^E)\|_1))$. We have*

$$\mathbb{E}\left(\sum_{n=0}^{H-1} \|\rho_n^E - \rho_n^{EA}\|_1\right) \leq H^2 \delta^{BC}, \tag{2}$$

*and*

$$\mathbb{E}\left(\sum_{n=0}^{H-1}\|\mu_n^E - \mu_n^{EA}\|_1\right) \le H^2 \delta^{BC}. \tag{3}$$

*Proof.*

$$\mathbb{E}\big(\|\rho_n^E - \rho_n^{EA}\|_1\big) = \mathbb{E}\left(\sum_{x'}|\rho_n^E(x') - \rho_n^{EA}(x')|\right)$$

$$= \mathbb{E}\left(\sum_{x'}\left|\sum_{x,a}\rho_{n-1}^E(x)\,\pi_{n-1}^E(a|x,\rho_{n-1}^E)\,P(x'|x,a,\rho_{n-1}^E,\varepsilon_{n+1}^0) - \rho_{n-1}^{EA}(x)\,\pi_{n-1}^A(a|x,\rho_{n-1}^E)\,P(x'|x,a,\rho_{n-1}^E,\varepsilon_{n+1}^0)\right|\right)$$

$$= \mathbb{E}\left(\sum_{x'}\left|\sum_{x,a}P(x'|x,a,\rho_{n-1}^E,\varepsilon_{n+1}^0)\big(\rho_{n-1}^E(x)\pi_{n-1}^E(a|x,\rho_{n-1}^E) - \rho_{n-1}^{EA}(x)\pi_{n-1}^A(a|x,\rho_{n-1}^E)\big)\right|\right)$$

$$\le \mathbb{E}\left(\sum_{x'}\sum_{x,a}P(x'|x,a,\rho_{n-1}^E,\varepsilon_{n+1}^0)\,\big|\rho_{n-1}^E(x)\pi_{n-1}^E(a|x,\rho_{n-1}^E) - \rho_{n-1}^{EA}(x)\pi_{n-1}^A(a|x,\rho_{n-1}^E)\big|\right)$$

$$= \mathbb{E}\left(\sum_{x,a}\big|\rho_{n-1}^E(x)\pi_{n-1}^E(a|x,\rho_{n-1}^E) - \rho_{n-1}^{EA}(x)\pi_{n-1}^A(a|x,\rho_{n-1}^E)\big|\underbrace{\sum_{x'}P(x'|x,a,\rho_{n-1}^E,\varepsilon_{n+1}^0)}_{=1}\right)$$

$$\le \mathbb{E}\left(\sum_{x,a}\big|\rho_{n-1}^E(x)\big(\pi_{n-1}^E(a|x,\rho_{n-1}^E) - \pi_{n-1}^A(a|x,\rho_{n-1}^E)\big) + \big(\rho_{n-1}^E(x) - \rho_{n-1}^{EA}(x)\big)\pi_{n-1}^A(a|x,\rho_{n-1}^E)\big|\right)$$

$$\le \mathbb{E}\left(\sum_{x,a}\rho_{n-1}^E(x)\,|\pi_{n-1}^E(a|x,\rho_{n-1}^E) - \pi_{n-1}^A(a|x,\rho_{n-1}^E)|\right) + \mathbb{E}\left(\sum_{x,a}|\rho_{n-1}^E(x) - \rho_{n-1}^{EA}(x)|\,\pi_{n-1}^A(a|x,\rho_{n-1}^E)\right)$$

$$= \underbrace{\mathbb{E}\left(\mathbb{E}_{x\sim\rho_{n-1}^E}\big[|\pi_{n-1}^E(x,\rho_{n-1}^E) - \pi_{n-1}^A(x,\rho_{n-1}^E)|\big]\right)}_{=\,\delta_{n-1}^{BC}} + \mathbb{E}\left(\sum_{x}|\rho_{n-1}^E(x) - \rho_{n-1}^{EA}(x)|\underbrace{\sum_{a}\pi_{n-1}^A(a|x,\rho_{n-1}^E)}_{=1}\right)$$

$$\le \delta_{n-1}^{BC} + \mathbb{E}\big(\|\rho_{n-1}^E - \rho_{n-1}^{EA}\|_1\big)$$

$$\le n\,\delta^{BC}.$$

That gives Eq. (2). Now,

$$\mathbb{E}\big(\|\mu_n^E - \mu_n^{EA}\|_1\big) = \mathbb{E}\left(\sum_{x,a}|\mu_n^E(x,a) - \mu_n^{EA}(x,a)|\right)$$

$$= \mathbb{E}\left(\sum_{x,a}|\rho_n^E(x)\,\pi_n^E(a|x,\rho_{n-1}^E) - \rho_n^{EA}(x)\,\pi_n^A(a|x,\rho_{n-1}^E)|\right)$$

$$= \mathbb{E}\left(\sum_{x,a}|\rho_n^E(x)\big(\pi_n^E(a|x,\rho_{n-1}^E) - \pi_n^A(a|x,\rho_{n-1}^E)\big) + \pi_n^A(a|x,\rho_{n-1}^E)\big(\rho_n^E(x) - \rho_n^{EA}(x)\big)|\right)$$

$$\le \mathbb{E}\left(\sum_{x,a}\rho_n^E(x)|\pi_n^E(a|x,\rho_{n-1}^E) - \pi_n^A(a|x,\rho_{n-1}^E)|\right) + \mathbb{E}\left(\sum_{x,a}\pi_n^A(a|x,\rho_{n-1}^E)|\rho_n^E(x) - \rho_n^{EA}(x)|\right)$$

$$\leq \delta_{n-1}^{BC} + \underbrace{\mathbb{E}\big(\|\rho_n^E - \rho_n^{EA}\|_1\big)}_{\leq n\,\delta^{BC}}$$

$$\leq (n+1)\,\delta^{BC}.$$

That gives Eq. (3). $\qquad\qquad\qquad\qquad\qquad\qquad\qquad\qquad\qquad\qquad\qquad\qquad\qquad\qquad\qquad\quad\square$

We extend (Ramponi et al., 2023, Lemma 1 C.1).

**Lemma A.3.** *Under Assumption* (2.3) (2.4)*, for any policies $\pi^A, \pi^1 \in \Pi$, we have*

$$
\begin{aligned}
V(\pi^1, \pi^A) - V(\pi^A, \pi^A) \leq & 2L_r\,\mathbb{E}\Big(\sum_{n=0}^{H-1}\|\rho_n^A - \rho_n^E\|_1\Big) \\
& + r_{\max}\Big[\mathbb{E}\Big(\sum_{n=0}^{H-1}\|\mu_n^E - \mu_n^{E,A}\|\Big) + \mathbb{E}\Big(\sum_{n=0}^{H-1}\|\rho_n^{A,1} - \rho_n^{E,1}\|_1\Big) + \mathbb{E}\Big(\sum_{n=0}^{H-1}\|\rho_n^{A,A} - \rho_n^{E,A}\|_1\Big) \\
& + \sum_{n=0}^{H-1}\mathbb{E}\Big(\mathbb{E}_{X\sim\rho_n^{E,1}}\big(\|\pi^1(X, \rho_n^A) - \pi^1(X, \rho_n^E)\|\big)\Big) + \sum_{n=0}^{H-1}\mathbb{E}\Big(\mathbb{E}_{X\sim\rho_n^{E,A}}\big(\|\pi^A(X, \rho_n^A) - \pi^A(X, \rho_n^E)\|\big)\Big)\Big].
\end{aligned}
\tag{4}
$$

*And by using the Lipschitz property of our policies:*

$$
\begin{aligned}
V(\pi^1, \pi^A) - V(\pi^A, \pi^A) \leq & 2(L_r + L_E\,r_{\max})\,\mathbb{E}\Big(\sum_{n=0}^{H-1}\|\rho_n^A - \rho_n^E\|_1\Big) \\
& + r_{\max}\Big[\mathbb{E}\Big(\sum_{n=0}^{H-1}\|\mu_n^E - \mu_n^{E,A}\|_1\Big) + \mathbb{E}\Big(\sum_{n=0}^{H-1}\|\rho_n^{A,1} - \rho_n^{E,1}\|_1\Big) + \mathbb{E}\Big(\sum_{n=0}^{H-1}\|\rho_n^{A,A} - \rho_n^{E,A}\|_1\Big)
\end{aligned}
\tag{5}
$$

*Proof.* Let $\pi^1 \in \Pi$, we can write

$$V(\pi^1, \pi^A) - V(\pi^A, \pi^A) = \underbrace{V(\pi^1, \pi^A) - V(\pi^E, \pi^E)}_{A} + \underbrace{V(\pi^E, \pi^E) - V(\pi^A, \pi^A)}_{B}.$$

First, let us work on $A$:

$$A = \underbrace{V(\pi^1, \pi^A) - V(\pi^1, \pi^E)}_{A_1} + \underbrace{V(\pi^1, \pi^E) - V(\pi^E, \pi^E)}_{A_2}.$$

$A_2$ is non positive since $\pi^E$ is an equilibrium. We bound $A_1$ with Lemma A.1 and obtain

$$A \leq L_r\,\mathbb{E}\Big(\sum_{n=0}^{H-1}\|\rho_n^A - \rho_n^E\|_1\Big) + r_{\max}\mathbb{E}\Big(\sum_{n=0}^{H-1}\|\rho_n^{A,1} - \rho_n^{E,1}\|_1\Big) + r_{\max}\sum_{n=0}^{H-1}\mathbb{E}\big(\mathbb{E}_{X\sim\rho_n^{E,1}}(\|\pi_n^1(X, \rho_n^A) - \pi_n^1(X, \rho_n^E)\|)\big).$$

Consider $B$:

$$B \leq \underbrace{V(\pi^E, \pi^E) - V(\pi^A, \pi^E)}_{B_1} + \underbrace{V(\pi^A, \pi^E) - V(\pi^A, \pi^A)}_{B_2}$$

$$
\begin{aligned}
B_1 &\leq \mathbb{E}\left(\sum_{n=0}^{H-1}\sum_{x,a}\big(\mu_n^{E,E}(x,a)r(x,a,\rho^E) - \mu_n^{E,A}(x,a)r(x,a,\rho^E)\big)\right) \\
&\leq r_{\max}\mathbb{E}\left(\sum_{n=0}^{H-1}\sum_{x,a}\big|\mu_n^{E,E}(x,a) - \mu_n^{E,A}(x,a)\big|\right)
\end{aligned}
$$

$$\leq r_{\max} \mathbb{E}\Big(\sum_{n=0}^{H-1} \|\mu_n^{E,E} - \mu_n^{E,A}\|\Big).$$

By Lemma A.1 we obtain

$$B_2 \leq L_r \, \mathbb{E}\Big(\sum_{n=0}^{H-1} \|\rho_n^A - \rho_n^E\|_1\Big) + r_{\max} \mathbb{E}\Big(\sum_{n=0}^{H-1} \|\rho_n^{A,A} - \rho_n^{E,A}\|_1\Big) + r_{\max} \sum_{n=0}^{H-1} \mathbb{E}\big(\mathbb{E}_{x \sim \rho^{E,A}}(\|\pi_n^A(x, \rho_n^A) - \pi_n^A(x, \rho_n^E))\big).$$

Summing everything we obtain Eq. (4). $\qquad\square$

We give two new lemmas.

**Lemma A.4.**

$$\mathbb{E}\Big(\sum_{n=0}^{H-1} \|\rho_n^A - \rho_n^E\|_1\Big) \leq \delta^{BC} \frac{(1 + L_E + L_P)^H}{(L_E + L_P)^2}, \tag{6}$$

$$\mathbb{E}(\|\rho_n^A - \rho_n^E\|_1) \leq \mathbb{E}(\|\mu_n^A - \mu_n^E\|_1) \leq \delta^{ADV}. \tag{7}$$

*Proof.*

$$\begin{aligned}
\mathbb{E}(\|\rho_{n+1}^A - \rho_{n+1}^E\|) &= \mathbb{E}\Big(\sum_{x',x,a} |\pi_n^A(a|x,\rho_n^A)\rho_n^A(x)P(x'|x,a,\rho_n^A,\varepsilon_{n+1}^0) - \pi_n^E(a|x,\rho_n^E)\rho_n^E(x)P(x'|x,a,\rho_n^E\varepsilon_{n+1}^0)|\Big) \\
&\leq \mathbb{E}\Big(\sum_{x',x,a} \pi_n^A(a|x,\rho_n^A)\rho_n^A(x)|P(x'|x,a,\rho_n^A,\varepsilon_{n+1}^0) - P(x'|x,a,\rho_n^E,\varepsilon_{n+1}^0)|\Big) \\
&\quad + \mathbb{E}\Big(\sum_{x',x,a} P(x'|x,a,\rho_n^E,\varepsilon_{n+1}^0)|\pi_n^A(a|x,\rho_n^A)\rho_n^A(x) - \pi_n^E(a|x,\rho_n^E)\rho_n^E(x)|\Big) \\
&\leq \underbrace{\mathbb{E}\Big(\sum_{x,a} \pi_n^A(a|x,\rho_n^A)\rho_n^A(x)\|P(x,a,\rho_n^A,\varepsilon_{n+1}^0) - P(x,a,\rho_n^E,\varepsilon_{n+1}^0)\|_1\Big)}_{\mathbb{E}(\mathbb{E}_{(x,a)\sim\mu_n^A}(\|P(x,a,\rho_n^A,\varepsilon_{n+1}^0)-P(x,a,\rho_n^E,\varepsilon_{n+1}^0)\|_1))} \\
&\quad + \mathbb{E}\Big(\sum_{x,a} |\pi_n^A(a|x,\rho_n^A)\rho_n^A(x) - \pi_n^E(a|x,\rho_n^E)\rho_n^E(x)|\Big) \\
&\leq L_P \mathbb{E}(\|\rho_n^A - \rho_n^E\|_1) + \underbrace{\mathbb{E}\Big(\sum_{x,a} |\pi_n^A(a|x,\rho_n^A)\rho_n^A(x) - \pi_n^A(a|x,\rho_n^A)\rho_n^E(x)|\Big)}_{A_1} \\
&\quad + \underbrace{\mathbb{E}\Big(\sum_{x,a} |\pi_n^A(a|x,\rho_n^A)\rho_n^E(x) - \pi_n^A(a|x,\rho_n^E)\rho_n^E(x)|\Big)}_{A_2} \\
&\quad + \underbrace{\mathbb{E}\Big(\sum_{x,a} |\pi_n^A(a|x,\rho_n^E)\rho_n^E(x) - \pi_n^E(a|x,\rho_n^E)\rho_n^E(x)|\Big)}_{A_3}
\end{aligned}$$

First we have $A_1 = \mathbb{E}(\|\rho_n^A - \rho_n^E\|_1)$. Then we have

$$A_2 = \mathbb{E}\big(\mathbb{E}_{x\sim\rho_n^E}(\|\pi_n^A(x,\rho_n^A) - \pi_n^A(x,\rho_n^E)\|_1)\big) \leq L_E \, \mathbb{E}(\|\rho_n^A - \rho_n^E\|_1).$$

Now,

$$A_3 = \underbrace{\mathbb{E}\big(\mathbb{E}_{x\sim\rho_n^E}(\|\pi_n^A(x,\rho_n^E) - \pi_n^E(x,\rho_n^E)\|)\big)}_{\leq \delta^{BC}}.$$

By combining the terms we have

$$\mathbb{E}(\|\rho_{n+1}^A - \rho_{n+1}^E\|) \leq \mathbb{E}(\|\rho_n^A - \rho_n^E\|)(1 + L_E + L_P) + \delta^{BC} \leq \delta^{BC} \sum_{k=0}^n (1 + L_E + L_P)^k = \delta^{BC} \frac{(1 + L_E + L_P)^n - 1}{L_E + L_P}.$$

By summing over $n$ we obtain the result. For Eq. (7), we have

$$\begin{aligned}
\mathbb{E}(\|\rho_n^A - \rho_n^E\|_1) &\leq \mathbb{E}(\sum_x |\rho_n^A(x) - \rho_n^E(x)|) \\
&\leq \mathbb{E}(\sum_x |\sum_a \mu_n^A(x,a) - \mu_n^E(x,a)|) \\
&\leq \mathbb{E}(\sum_{x,a} |\mu_n^A(x,a) - \mu_n^E(x,a)|) \\
&\leq \mathbb{E}(\|\mu_n^A - \mu_n^E\|_1) \leq \delta^{ADV}.
\end{aligned}$$

$\square$

**Lemma A.5.** *Let $\pi^1 \in \Pi$. We have*

$$\mathbb{E}(\sum_{n=0}^{H-1} \|\rho_n^{A,1} - \rho_n^{E,1}\|) \leq \delta^{BC} \frac{(1 + L_E + L_P)^H}{(L_P + L_E)^2}. \tag{8}$$

*and*

$$\mathbb{E}(\sum_{n=0}^{H-1} \|\rho_n^{A,1} - \rho_n^{E,1}\|) \leq \delta^{ADV}(L_P + L_E)H^2. \tag{9}$$

*Proof.* Suppose $L_P + L_E \neq 0$,

$$\begin{aligned}
\mathbb{E}(\|\rho_n^{A,1} - \rho_n^{E,1}\|) &= \mathbb{E}\left( \sum_{x'} |\rho_n^{A,1}(x') - \rho_n^{E,1}(x')| \right) \\
&\leq \mathbb{E}\left( \sum_{x',x,a} |\rho_{n-1}^{A,1}(x)\pi_{n-1}^1(a|x,\rho_{n-1}^A)P(x'|x,a,\rho_{n-1}^A\varepsilon_n^0) - \rho_{n-1}^{E,1}(x)\pi_{n-1}^1(a|x,\rho_{n-1}^E)P(x'|x,a,\rho_{n-1}^E\varepsilon_n^0)| \right) \\
&\leq \mathbb{E}\left( \sum_{x',x,a} \rho_{n-1}^{A,1}(x)\pi_{n-1}^1(a|x,\rho_{n-1}^A)|P(x'|x,a,\rho_{n-1}^A,\varepsilon_n^0) - P(x'|x,a,\rho_{n-1}^E,\varepsilon_n^0)| \right) \\
&\quad + \mathbb{E}\left( \sum_{x',x,a} P(x'|x,a,\rho_{n-1}^E,\varepsilon_{n+1}^0)|\rho_{n-1}^{A,1}(x)\pi_{n-1}^1(a|x,\rho_{n-1}^A) - \rho_{n-1}^{E,1}(x)\pi_{n-1}^1(a|x,\rho_{n-1}^E)| \right) \\
&\leq \underbrace{\mathbb{E}\left( \sum_{x,a} \rho_{n-1}^{A,1}(x)\pi_{n-1}^1(a|x,\rho_{n-1}^A)\|P(x,a,\rho_{n-1}^A,\varepsilon_n^0) - P(x,a,\rho_{n-1}^E,\varepsilon_n^0)\|_1 \right)}_{A_1} \\
&\quad + \underbrace{\mathbb{E}\left( \sum_{x,a} |\rho_{n-1}^{A,1}(x)\pi_{n-1}^1(a|x,\rho_{n-1}^A) - \rho_{n-1}^{E,1}(x)\pi_{n-1}^1(a|x,\rho_{n-1}^E)| \right)}_{A_2}
\end{aligned}$$

Let us consider $A_1$

$$A_1 = \mathbb{E}\left( \mathbb{E}_{(x,a)\sim\mu_{n-1}^{A,1}}(\|P(x,a,\rho_{n-1}^A,\varepsilon_n^0) - P(x,a,\rho_{n-1}^E,\varepsilon_n^0)\|_1) \right) \leq L_P\mathbb{E}(\|\rho_{n-1}^A - \rho_{n-1}^E\|).$$

Let us consider $A_2$

$$A_2 = \mathbb{E}\left(\sum_{x,a} |\rho_{n-1}^{A,1}(x)\pi_{n-1}^1(a|x,\rho_{n-1}^A) - \rho_{n-1}^{E,1}(x)\pi_{n-1}^1(a|x,\rho_{n-1}^E)|\right)$$

$$\leq \mathbb{E}\left(\sum_{x,a} |\pi_{n-1}^1(a|x,\rho_{n-1}^A)\left(\rho_{n-1}^{A,1}(x) - \rho_{n-1}^{E,1}(x)\right) + \rho_{n-1}^{E,1}(x)\left(\pi_{n-1}^1(a|x,\rho_{n-1}^E) - \pi_{n-1}^1(a|x,\rho_{n-1}^A)\right)|\right)$$

$$\leq \mathbb{E}\left(\sum_{x,a} \pi_{n-1}^1(a|x,\rho_{n-1}^A)|\rho_{n-1}^{A,1}(x) - \rho_{n-1}^{E,1}(x)|\right) + \mathbb{E}\left(\sum_{x,a} \rho_{n-1}^{E,1}(x)|\pi_{n-1}^1(a|x,\rho_{n-1}^E) - \pi_{n-1}^1(a|x,\rho_{n-1}^A)|\right)$$

$$\leq \mathbb{E}(\|\rho_{n-1}^{A,1} - \rho_{n-1}^{E,1}\|) + \mathbb{E}(\mathbb{E}_{x\sim\rho^{E,1}}(\|\pi_{n-1}^1(x,\rho_{n-1}^E) - \pi_{n-1}^1(x,\rho_{n-1}^A)\|_1)$$

$$\leq \mathbb{E}(\|\rho_{n-1}^{A,1} - \rho_{n-1}^{E,1}\|) + L_E\mathbb{E}(\|\rho_{n-1}^E - \rho_{n-1}^A\|_1)$$

Combining $A_1$ and $A_2$ we obtain

$$\mathbb{E}(\|\rho_n^{A,1} - \rho_n^{E,1}\|) \leq \mathbb{E}(\|\rho_{n-1}^{A,1} - \rho_{n-1}^{E,1}\|) + (L_E + L_P)\mathbb{E}(\|\rho_{n-1}^E - \rho_{n-1}^A\|_1)$$

$$\leq \mathbb{E}(\|\rho_{n-1}^{A,1} - \rho_{n-1}^{E,1}\|) + \delta^{BC}(L_E + L_P)\frac{(1 + L_E + L_P)^{n-1} - 1}{L_E + L_P}$$

$$\leq \mathbb{E}(\|\rho_{n-1}^{A,1} - \rho_{n-1}^{E,1}\|) + \delta^{BC}(1 + L_E + L_P)^{n-1}$$

By summing we obtain

$$\mathbb{E}(\|\rho_n^{A,1} - \rho_n^{E,1}\|) \leq \delta^{BC}\frac{(1 + L_E + L_P)^n}{L_P + L_E}$$

and by summing again

$$\mathbb{E}(\sum_{n=0}^{H-1} \|\rho_n^{A,1} - \rho_n^{E,1}\|) \leq \delta^{BC}\frac{(1 + L_E + L_P)^n}{(L_P + L_E)^2}.$$

Let us prove Eq. (9). Previously, one of the intermediate step showed that $\mathbb{E}(\|\rho_n^{A,1} - \rho_n^{E,1}\|) \leq \mathbb{E}(\|\rho_{n-1}^{A,1} - \rho_{n-1}^{E,1}\|) + (L_P + L_E)\mathbb{E}(\|\rho_{n-1}^E - \rho_{n-1}^A\|_1)$. Applying Lemma A.4 with Eq. (7) leads to

$$\mathbb{E}(\|\rho_n^{A,1} - \rho_n^{E,1}\|) \leq \mathbb{E}(\|\rho_{n-1}^{A,1} - \rho_{n-1}^{E,1}\|) + (L_E + L_P)\delta^{ADV},$$

from what we deduce the result by summing twice. $\square$

**Lemma A.6.** *We have*

$$\mathbb{E}(\sum_{n=1}^{H-1} \|\mu_n^{EA} - \mu_n^E\|_1) \leq \delta^{ADV}(H(L_E + 1) + H^2(L_P + L_E)).$$

*Proof.*

$$\mathbb{E}(|\mu_n^{EA} - \mu_n^E\|_1) \leq \mathbb{E}(\|\mu_n^{EA} - \mu_n^A\|_1) + \underbrace{\mathbb{E}(\|\mu_n^A - \mu_n^E\|_1)}_{\leq \delta^{ADV}}.$$

Now we have

$$\mathbb{E}(\|\mu_n^{EA} - \mu_n^A\|_1) \leq \mathbb{E}(\sum_{x,a} |\rho_n^{EA}(x)\pi_n^A(a|x,\rho_n^E) - \rho_n^A(x)\pi_n^A(a|x,\rho_n^A)|)$$

$$\leq \underbrace{\mathbb{E}(\sum_{x,a} \rho_n^{EA}(x)|\pi_n^A(a|x,\rho^E) - \pi_n^A(a|x,\rho_n^A)|)}_{= \mathbb{E}\left(\mathbb{E}_{X\sim\rho_n^{EA}}(\|\pi^A(X,\rho_n^E) - \pi^A(X,\rho_n^A)\|)\right)} + \mathbb{E}(\sum_{x,a} \pi_n^A(a|x,\rho_n^A)|\rho_n^{EA}(x) - \rho_n^A(x)|)$$

$$\leq L_E\mathbb{E}(\|\rho_n^E - \rho_n^A\|_1) + \mathbb{E}(\|\rho_n^{EA} - \rho_n^A\|_1).$$

We deduce that

$$\mathbb{E}(|\mu_n^{EA} - \mu_n^E\|_1) \leq L_E \mathbb{E}(\|\rho_n^E - \rho_n^A\|_1) + \mathbb{E}(\|\rho_n^{EA} - \rho_n^A\|_1) + \delta^{ADV}.$$

Applying Lemma A.4 with Eq. (7) and Lemma A.5 with Eq. (9) we obtain

$$\mathbb{E}(\sum_{n=1}^{H-1} \|\mu_n^{EA} - \mu_n^E\|_1) \leq \delta^{ADV}(H(L_E + 1) + H^2(L_P + L_E)),$$

□

## A.2. Proofs of the Theorems

To prove the theorems, it is enough to apply the previous established inequalities.

**Proof of Theorem 3.1**

*Proof.* Apply Lemma A.3 with Eq. (5), then Lemma A.2, Lemma A.4 with Eq. (6) and Lemma A.5 with Eq. (8). □

**Proof of Theorem 3.2**

*Proof.* Apply Lemma A.3 with Eq. (5), then Lemma A.6, Lemma A.4 with Eq. (7) and Lemma A.5 with Eq. (9). □

**Proof of Theorem 3.5**

*Proof.* In the proof of Lemma A.3, we show that

$$|V(\pi^E, \pi^E) - V(\pi^A, \pi^E)| \leq r_{\max} \mathbb{E}\left[\sum_{n=0}^{H-1} \|\mu_n^E - \mu^{E,A}\|_1\right].$$

We can conclude by applying Lemma A.2. □

**Proof of Theorem 3.6**

*Proof.* The proof uses the same argument as in Theorem 3.5, but by using Lemma A.6. □

# B. Additional Algorithms

---
**Algorithm 4** Proxy Algorithm
---

**Require:** MFG $\mathcal{M} = (\mathcal{X}, \mathcal{A}, \mathcal{E}^0, \nu^0, P, r, H, \rho_0)$;
$\qquad\quad \pi^A, \pi^E$;
$\qquad\quad$ Monte Carlo parameter $N$;

1: Initialize the empirical BC proxy for $t = 0, \ldots, H - 1$, $\hat{\delta}_t^{BC} = 0$.
2: Initialize the empirical ADV proxy for $t = 0, \ldots, H - 1$, $\hat{\delta}_t^{ADV} = 0$.
3: Initialize
$$\hat{\delta}_0^{BC} = \sum_x \rho_0(x) \| \pi_0^A(x, \rho_0) - \pi_0^E(x, \rho_0) \|_1.$$

4: Initialize
$$\hat{\delta}_0^{ADV} = \sum_{x,a} |\rho_0(x) \pi_0^A(a|x, \rho_0) - \rho_0(x) \pi_0^E(a|x, \rho_0)|.$$

5: **for** $n = 1, \ldots, N$ **do**
6: $\quad$ Sample one i.i.d sequence of common noises $\varepsilon_1^{0,n}, \ldots, \varepsilon_{H-1}^{0,n}$ with $\varepsilon_1^{0,n} \sim \nu^0$.
7: $\quad$ Initialize the mean-field $\rho_0^{E,n} = \rho_0$.
8: $\quad$ Initialize the mean-field $\rho_0^{A,n} = \rho_0$.
9: $\quad$ **for** $t = 1, \ldots, H - 1$ **do**
10: $\qquad$ Compute
$$\rho_t^{E,n} = \sum_{x,a} \rho_{t-1}^{E,n}(x) \pi_{t-1}^E(a \mid x, \rho_{t-1}^{E,n}) P(\cdot \mid x, a, \rho_{t-1}^{E,n}, \varepsilon_t^{0,n}).$$

11: $\qquad$ Compute
$$\rho_t^{A,n} = \sum_{x,a} \rho_{t-1}^{A,n}(x) \pi_{t-1}^{A,n}(a \mid x, \rho_{t-1}^{A,n}) P(\cdot \mid x, a, \rho_{t-1}^{A,n}, \varepsilon_t^{0,n}).$$

12: $\qquad$ Update
$$\hat{\delta}_t^{BC} \mathrel{+}= \frac{1}{N} \sum_x \rho_t^{E,n}(x) \| \pi_t^A(x, \rho_t^{E,n}) - \pi_t^E(x, \rho_t^{E,n}) \|_1.$$

13: $\qquad$ Update
$$\hat{\delta}_t^{ADV} \mathrel{+}= \frac{1}{N} \sum_{x,a} |\rho_t^{A,n}(x) \pi_t^A(a|x, \rho_t^{A,n}) - \rho_t^{E,n}(x) \pi_t^E(a|x, \rho_t^{E,n})|.$$

14: $\quad$ **end for**
15: **end for**
16: Compute $\hat{\delta}^{BC} = \max_{t \in [H-1]} \hat{\delta}_t^{BC}$.
17: Compute $\hat{\delta}^{ADV} = \max_{t \in [H-1]} \hat{\delta}_t^{ADV}$.
18: **Return** $\hat{\delta}^{BC}, \hat{\delta}^{ADV}$

---

---

**Algorithm 5** Neural Network Best Response

---

**Require:** MFG $\mathcal{M} = (\mathcal{X}, \mathcal{A}, \mathcal{E}^0, \nu^0, P, r, H, \rho_0)$; Sequence of policies $(\pi^0, \ldots, \pi^k) \in \Pi^{k+1}$; Iterations $J$; Batch size $B$; Optimizer.

1: **for** $j = 1, \ldots, J$ **do**
2:     Reset batch loss: $L(\theta_{j-1}) = 0$.
3:     Sample a batch of i.i.d. common noises $(\varepsilon_t^{0,s})_{0 \leq t \leq H-1}$ for $s = 1, \ldots, B$.
4:     **for** $s = 1, \ldots, B$ **do**
5:         Compute the aggregated sequence $\bar{\rho}^{k,s}$ induced by $(\pi^0, \ldots, \pi^k)$.
6:         Compute the mean-field sequence $\rho^{\theta,s}$ induced by the learner $\pi_\theta$ responding to $\bar{\rho}^{k,s}$.
7:         Accumulate the loss:

$$L(\theta_{j-1}) \leftarrow L(\theta_{j-1}) - \frac{1}{B} \sum_{t=0}^{H-1} \sum_{x,a} \rho_t^{\theta,s}(x) \pi_t^\theta(a \mid x, \bar{\rho}_t^{k,s}) r(x, a, \bar{\rho}_t^{k,s})$$

8:     **end for**
9:     Compute the gradient $\nabla_\theta L(\theta_{j-1})$ and update parameters via the optimizer to obtain $\theta_j$.
10: **end for**
11: **Return** Optimized parameters $\theta_J$ defining the best response policy $\pi_{\theta_J}$.

---

## C. Additional Details to Example 1

### C.1. Mann Iterations and Best Response

We use the following algorithm to compute a $\epsilon$-Nash equilibrium to obtain the expert policy $\pi^E$.

---
**Algorithm 6** Mann Iteration for $\epsilon$-Nash Equilibrium
---
**Require:** MFG environment $\mathcal{M}$; Initial policy $\pi^0 \in \Pi$, Learning rate sequence $(\gamma_k)_{1 \leq k \leq K} \in (0, 1]$; Number of iterations $K$.
1: Define $\pi^{0,*} = \pi^0$
2: **for** each $k = 0, \ldots, K - 1$ **do**
3:     Compute best response $\pi^{BR,k}$ to $\pi^{k-1,*}$.
4:     Define $\pi^{k,*} = \pi^{k-1,*} + \gamma_k \pi^{BR,k}$
5: **end for**
6: **Return** $\pi^{*,K}$.

---

To compute a best response, we discretize the mean-field space, and use a Monte Carlo estimation to perform a backward induction.

---
**Algorithm 7** Backward Induction for Best Response.
---
**Require:** Fixed population policy $\pi$; MFG $\mathcal{M}$; Grid for mean-field $\mathcal{G}_\rho$; Monte Carlo batch size $M$.
1:  **Initialize** Value function $V_H(x, \rho) = 0$ for all $x \in \mathcal{X}, \rho \in \mathcal{G}_\rho$.
2:  **for** time $t = H - 1, \ldots, 0$ **do**
3:     **for** each possible density $\rho \in \mathcal{G}_\rho$ **do**
4:         **for** each state $x \in \mathcal{X}$ **do**
5:             **for** each action $a \in \mathcal{A}$ **do**
6:                 Sample a batch of common noise $(\varepsilon^{0,s})_{1 \leq s \leq M}$.
7:                 **for** each $s \in [M]^*$ **do**
8:                     Compute expected next density: $\rho'^s = \Phi(\rho^s, \pi_t, \varepsilon_{t+1}^{0,s})$.
9:                 **end for**
10:                Define $Q_t(x, \rho, a) = r(x, a, \rho) + \sum_{s=1}^{M} V_{t+1}(x', \rho'^s) \rho'^s(x')$
11:             **end for**
12:         $V_t(x, \rho) = \max_{a \in \mathcal{A}} Q_t(x, \rho, a)$
13:         $\pi_t^{BR}(a \mid x, \rho) = \mathbb{1}_{\{a = \arg\max_{a'} Q_t(x, \rho, a')\}}$
14:         **end for**
15:     **end for**
16: **end for**
17: **Return** $\pi^{BR} = (\pi_t^{BR})_{0 \leq t < H}$.

---

### C.2. Vanilla and Adaptive Policies

We describe here how we obtain using data from expert agents following the policy $\pi^E$, the vanilla and adaptive policies used for the experiments in Subsection 4.6.

**Data Generation:** The simplicity of the environment allows us to consider classical statistical estimators for the vanilla and adaptive policies. Let $(\eta, \alpha)$ be a couple of parameters. We generate a synthetic dataset composed of $N$ independent trajectories of $M$ agents: $\mathcal{D}_{\alpha,\eta}^{N,M} = \left\{ \left(x_{t,\alpha,\eta}^{n,m}, a_{t,\alpha,\eta}^{n,m}\right)_{\substack{0 \leq t \leq H-1 \\ 0 \leq m \leq M-1}} \right\}_{1 \leq n \leq N}$, where each agent acts according to the expert policy $\pi_{\alpha,\eta}^E$ in an environment subjected to a common noise realization $\varepsilon_{1:H-1}^{0,n}$. Specifically, for each trajectory $n$, a unique common noise sequence $(\varepsilon_t^{0,n})_{0 \leq t \leq H-1}$ is sampled, inducing a specific mean-field sequence $\rho_{\alpha,\eta}^{E,n}(\varepsilon_{1:H-1}^{0,n})$ that the agents observe and react to. To simplify notation, we fix the parameters $(\alpha, \eta)$ and suppress their explicit dependence in the following subscripts.

**Vanilla Policy:** We define the vanilla policy $\hat{\pi}^{M,N,\text{vanilla}}$ as the time-dependent approximation that ignores the common noise fluctuations. It is computed via the empirical marginals:

$$\hat{\pi}_t^{M,N,\text{vanilla}}(a \mid x) = \frac{\hat{\mu}_t^{E,M,N}(x, a)}{\sum_{a' \in \mathcal{A}} \hat{\mu}_t^{E,M,N}(x, a')},$$

where $\hat{\mu}_t^{E,M,N}(x,a) = \frac{1}{NM} \sum_{n,m} \mathbb{1}_{\{x_t^{n,m}=x, a_t^{n,m}=a\}}$.

**Adaptive Policy:** We define the adaptive policy $\hat{\pi}^{M,N,\text{adaptive}}$ as a density-dependent policy. To reconstruct the policy for any arbitrary mean-field value $\rho$, we employ a Nadaraya–Watson kernel regression:

$$\hat{\pi}_t^{M,N,\text{adaptive}}(a \mid x, \rho) =$$

$$\frac{\sum_{n,m} \mathbb{1}_{\{x_t^{n,m}=x, a_t^{n,m}=a\}} K_h(\hat{\rho}_t^{E,n,M} - \rho)}{\sum_{a' \in \mathcal{A}} \sum_{n,m} \mathbb{1}_{\{x_t^{n,m}=x, a_t^{n,m}=a'\}} K_h(\hat{\rho}_t^{E,n,M} - \rho)},$$

where $\hat{\rho}_t^{E,n,M}(1) = \frac{1}{M} \sum_m \mathbb{1}_{\{x_t^{n,m}=1\}}$ is the empirical density of agents in state 1 for trajectory $n$. The kernel function $K_h : \Delta_{\mathcal{X}} \times \Delta_{\mathcal{X}} \to \mathbb{R}_+$ is defined by the Gaussian radial basis function:

$$K_h(\rho, \rho') = \exp\left(-\frac{\|\rho - \rho'\|_2^2}{2h^2}\right), \quad \forall (\rho, \rho') \in \Delta_{\mathcal{X}}^2.$$

This kernel density estimation (KDE) approach allows the agent to interpolate the expert's behavior across the continuous space of possible mean-field realizations.

### C.3. Hyperparameters

All the experiments have been run with parameters presented in Table 1.

*Table 1.* Summary of Hyperparameters for the First Example

| Category | Parameter | Symbol | Value |
|---|---|---|---|
| **Nash Equilibrium** | Mann Iterations | $K$ | 50 |
| | Learning Rate | $\gamma_k$ | 0.05 |
| | Mean Field Discretizations | $\mathcal{G}_\rho$ | 50 points |
| | MC Samples (Backward Induction) | - | 10000 |
| **Data Generation** | Number of Trajectories | $N$ | 2000 |
| | Number of Agents | $M$ | 100 |
| **Kernel Regression** | Bandwidth | $h$ | 0.05 |

### C.4. Full Sensitivity Analysis

Figure 12 represents the average over 5 runs of the metrics, as functions of the varying parameters $\eta$ and $\alpha$. Figure 13 shows the standard deviation. For the row corresponding to the proxy, the advantage corresponds to the adaptive proxy minus the vanilla proxy. For the rows corresponding to the reward vs expert, it corresponds to the reward of the adaptive policy minus the one of the vanilla policy. For the rows corresponding to the exploitability, it corresponds to the exploitability of the vanilla minus the one of the adaptive policy.

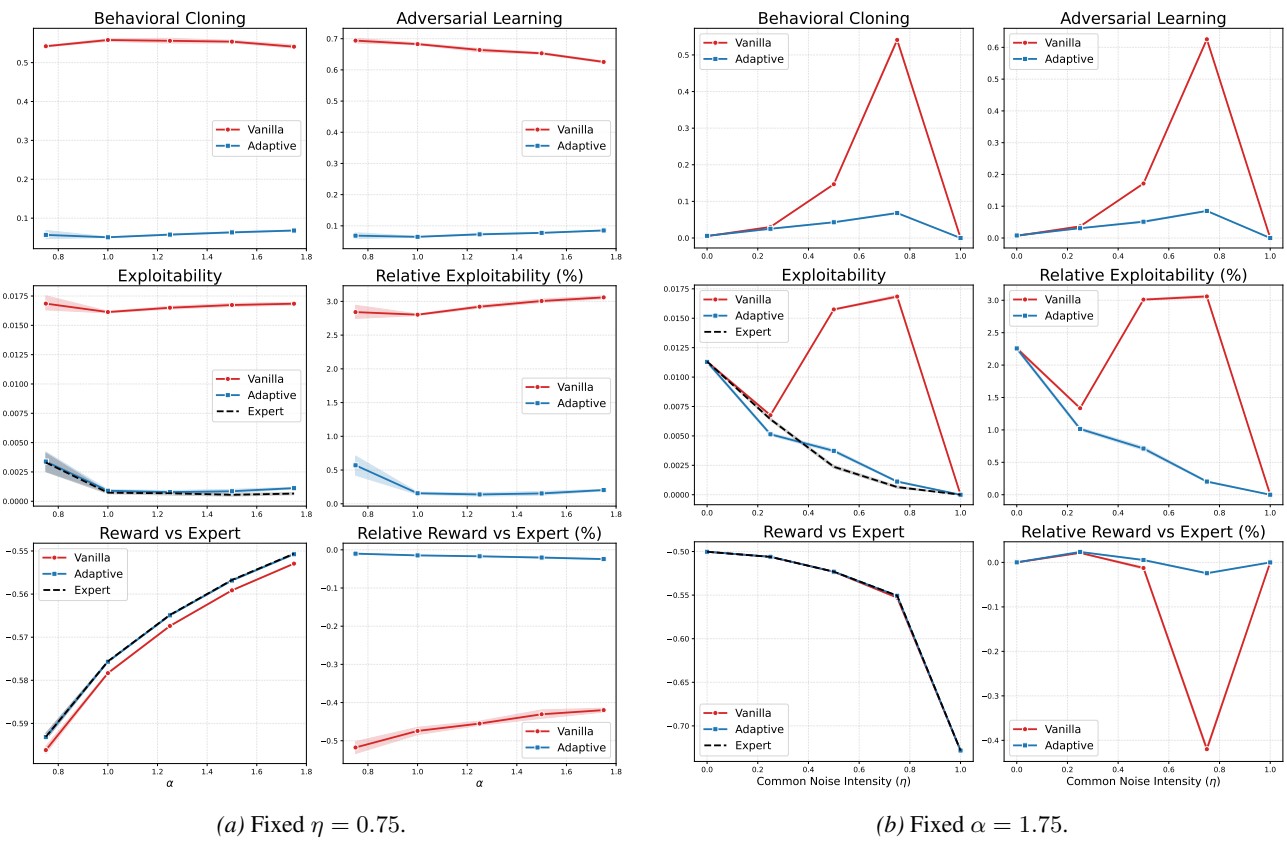

*(a)* Fixed $\eta = 0.75$.        *(b)* Fixed $\alpha = 1.75$.

*Figure 11.* Performance metrics for 5 runs in the Example 1. Left: variation across $\alpha$ with fixed $\eta$; Right: variation across $\eta$ with fixed $\alpha$.

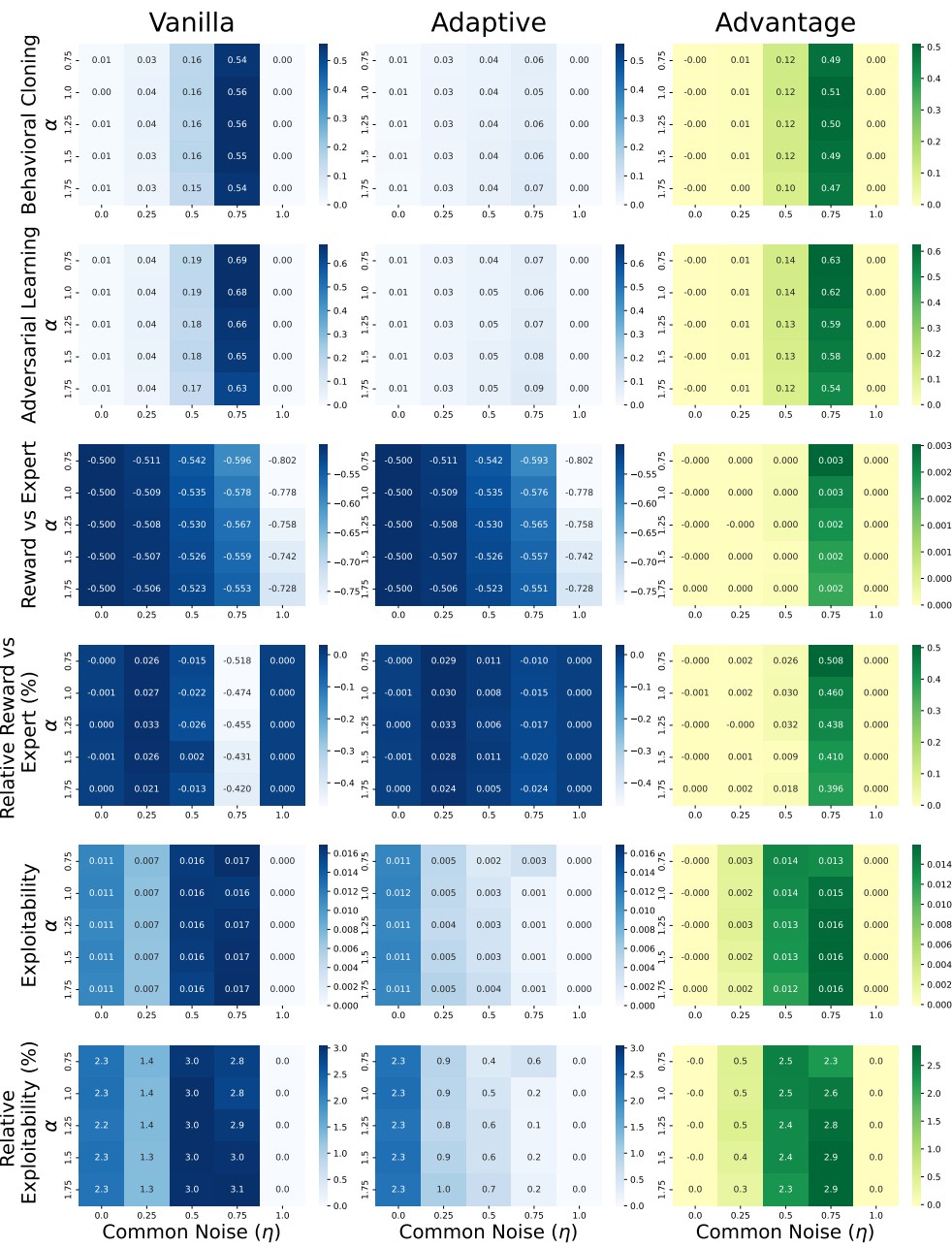

*Figure 12.* Example 1: Metrics Averages

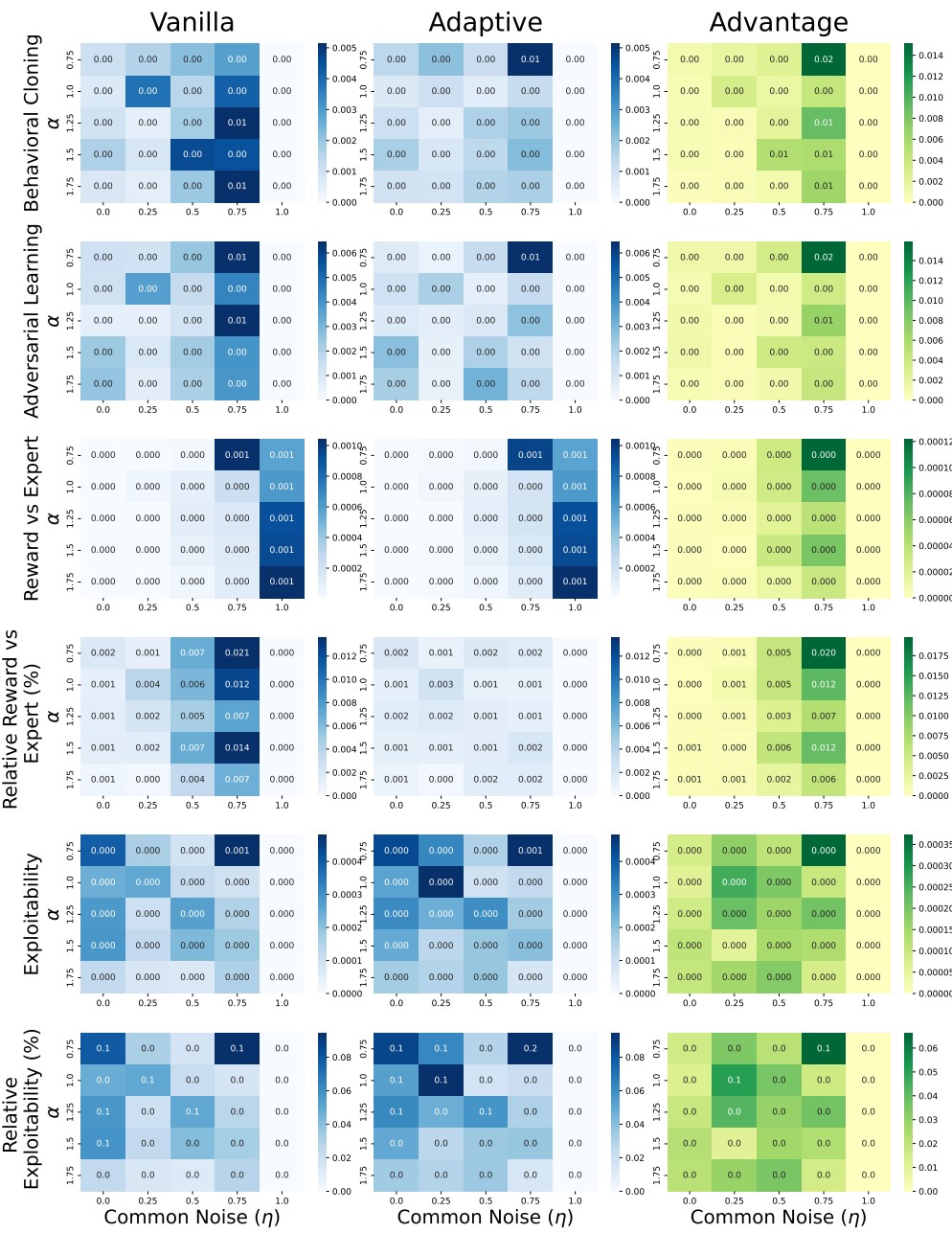

*Figure 13.* Example 1: Metrics STD

# D. Additional Details to the Beach Bar Example

## D.1. Hyperparameters

All the experiments have been run with the parameters presented in Table 2

*Table 2.* Summary of Hyperparameters for the Beach Bar Environment

| Category | Parameter | Value |
|---|---|---|
| **Neural Architecture** | Model Type | MLP |
| | Hidden Layers | 2 |
| | Layer Width | 64 |
| | Activation Function | ReLU |
| | Optimizer | Adam |
| **Fictitious Play** | Number of (FP) iterations | 40 |
| **Best Response Training** | Number of iterations | 1 000 |
| | Learning Rate | $10^{-4}$ |
| | Batch Size | 500 |
| **Expert Imitation** | Number of iterations | 10 000 |
| | Learning Rate | $10^{-4}$ |
| | Batch Size | 200 |
| **Vanilla and Adaptive Learning** | Number of iterations | 10 000 |
| | Learning Rate | $10^{-4}$ |
| | Batch Size | 50 |
| | Number of agents in the Simulation | 1000 |

## D.2. Full Sensitivity Analysis and Training Losses

Figure 15 represents the average over 5 runs of the metrics, as functions of the varying parameters $\eta$ and $\alpha$. Figure 16 shows the standard deviation. For the row corresponding to the proxy, the advantage corresponds to the adaptive proxy minus the vanilla proxy. For the rows corresponding to the reward vs expert, it corresponds to the reward of the adaptive policy minus the one of the vanilla policy. For the rows corresponding to the exploitability, it corresponds to the exploitability of the vanilla minus the one of the adaptive policy.

Figure 17 illustrates the convergence of Fictitious Play (FP) with common noise (Algorithm 1), the training loss of the expert learning to replicate the aggregated mean fields generated by the FP-outputted policies (Algorithm 2), and the training losses for the vanilla and adaptive parametric policies trained via Algorithm 3.

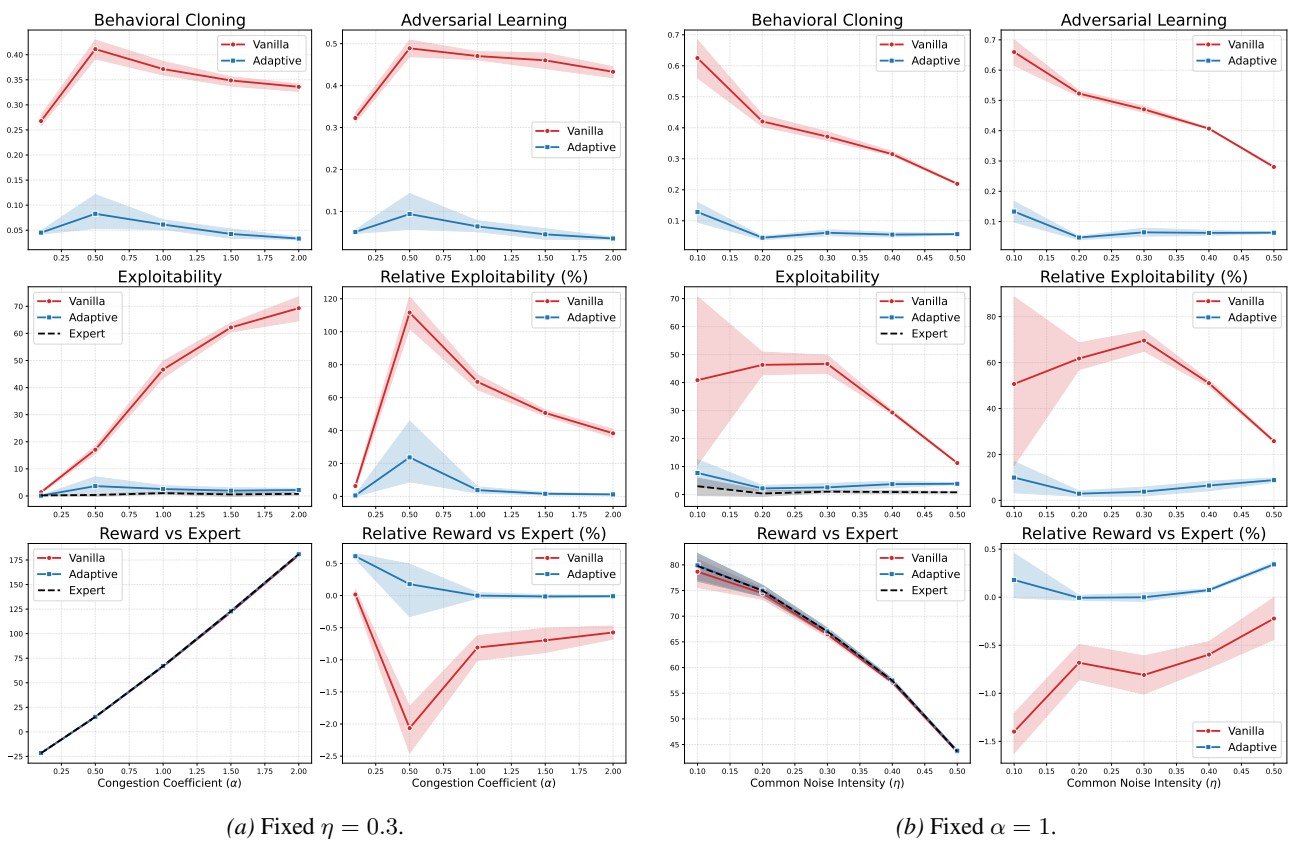

*(a)* Fixed $\eta = 0.3$.

*(b)* Fixed $\alpha = 1$.

*Figure 14.* Performance metrics for 5 runs in the Beach Bar environment. Left: variation across $\alpha$ with fixed $\eta$; Right: variation across $\eta$ with fixed $\alpha$.

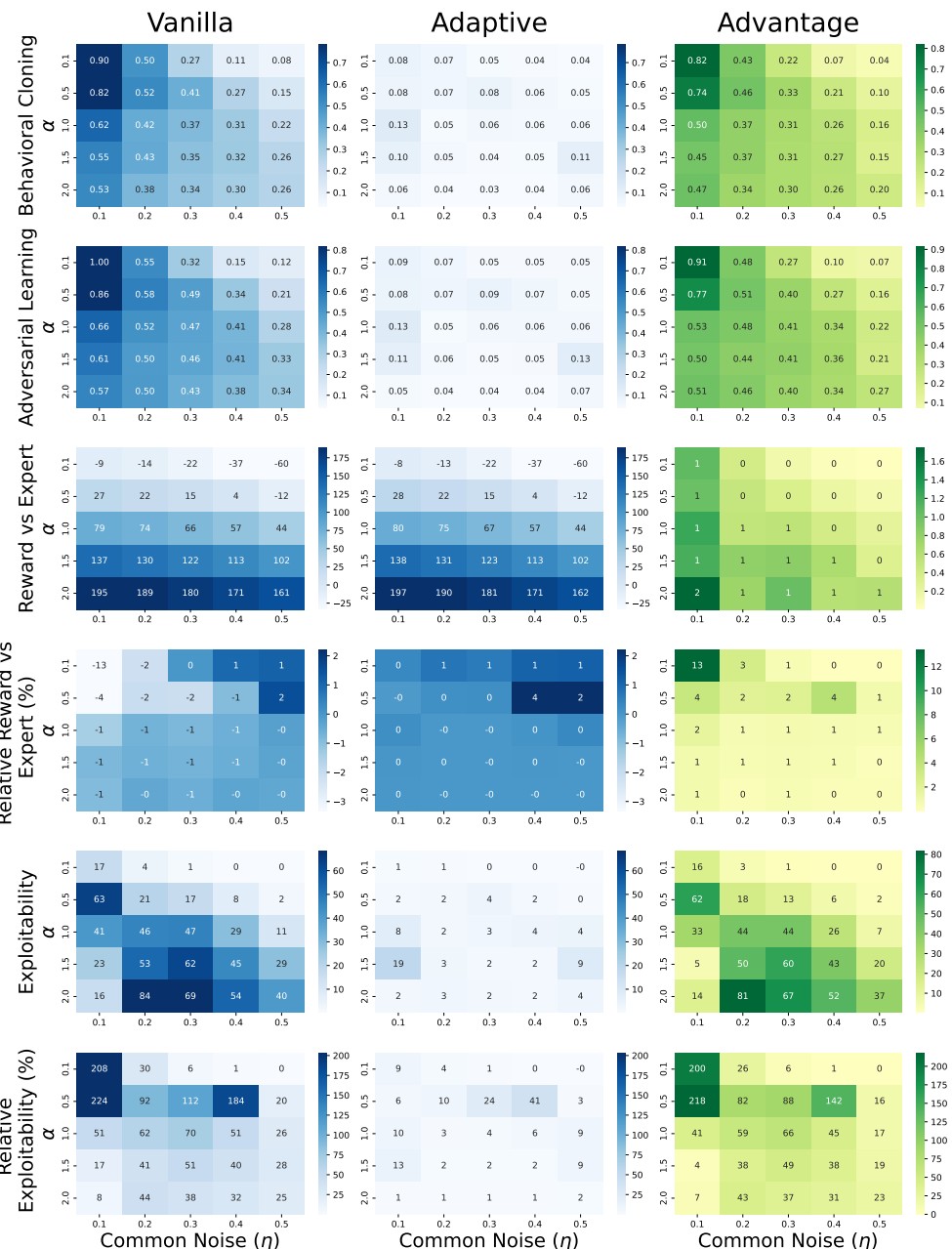

*Figure 15.* Beach Bar: Metrics Averages

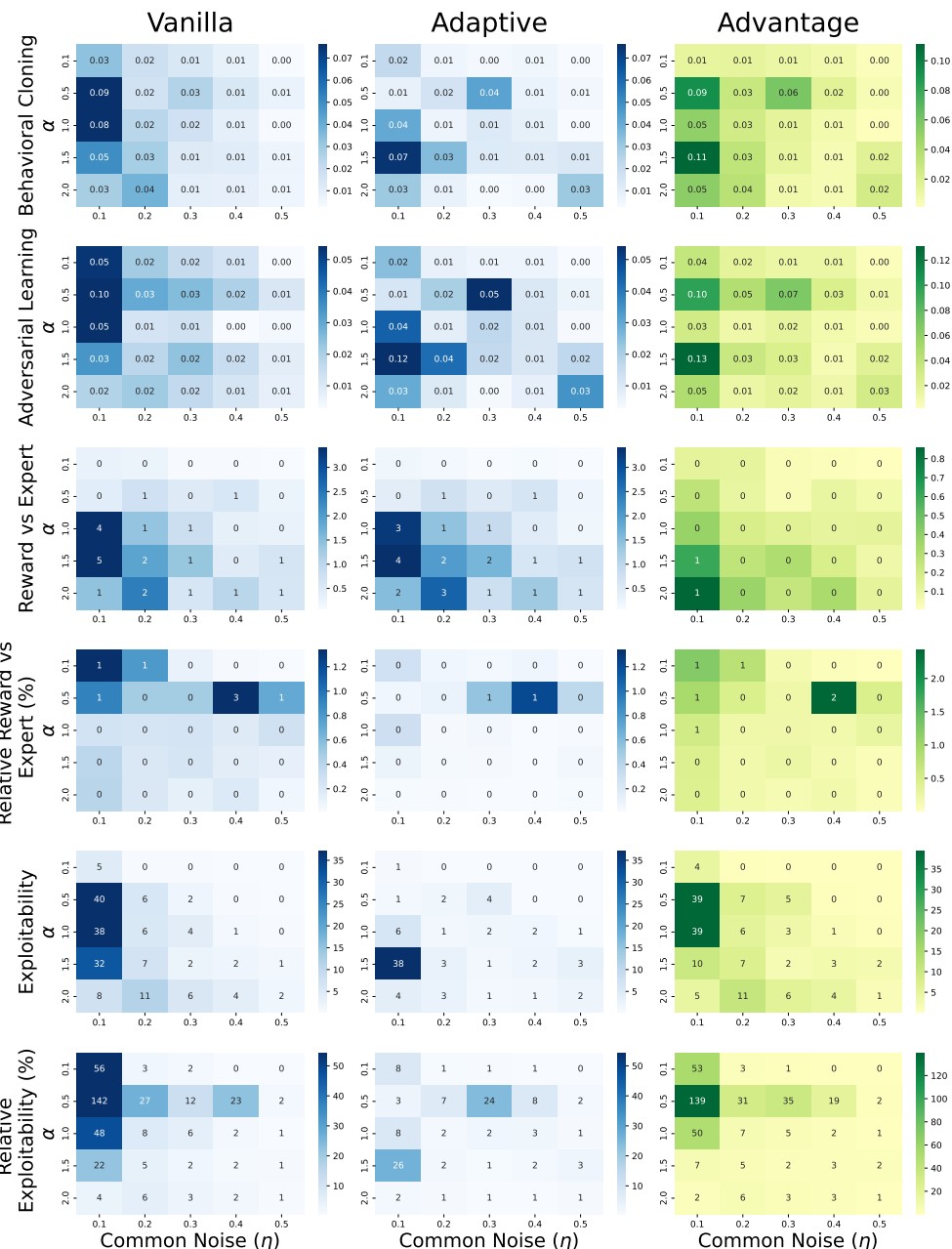

*Figure 16.* Beach Bar: Metrics STD.

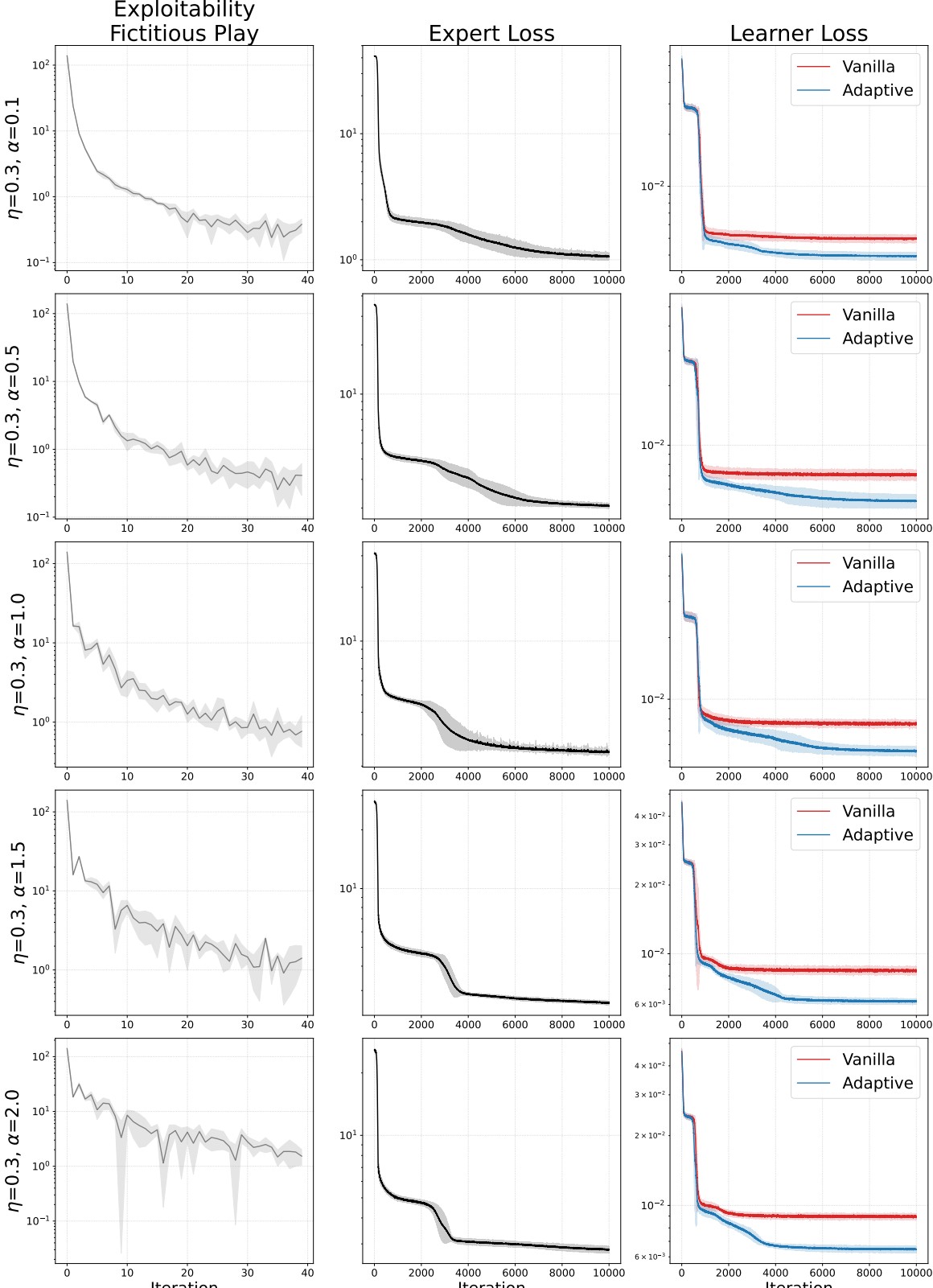

*Figure 17.* Beach Bar: Fictitious Play Exploitability Convergence, Expert Losses, Vanilla and Adaptive Losses ($\eta = 0.3$).

# E. Additional Details to the Night Clubs Example

## E.1. Hyperparameters

All the experiments have been ran with the parameters presented in Table 3

*Table 3.* Summary of Hyperparameters for the Night Club Environment

| Category | Parameter | Value |
|---|---|---|
| **Neural Architecture** | Model Type | MLP |
| | Hidden Layers | 2 |
| | Layer Width | 64 |
| | Activation Function | ReLU |
| | Optimizer | Adam |
| **Fictitious Play** | Number of (FP) iterations | 40 |
| **Best Response Training** | Number of iterations | 1 000 |
| | Learning Rate | $10^{-4}$ |
| | Batch Size | 500 |
| **Expert Imitation** | Number of iterations | 20 000 |
| | Learning Rate | $5 \times 10^{-4}$ |
| | Scheduler | Cosine Annealing |
| | Batch Size | 500 |
| **Vanilla and Adaptive Learning** | Number of iterations | 20 000 |
| | Learning Rate | $10^{-4}$ |
| | Batch Size | 50 |
| | Number of agents in the Simulation | 1000 |

## E.2. Full Sensitivity Analysis and Training Losses

Figure 19 represents the average over 5 runs of the metrics, as functions of the varying parameters $\eta$ and $\alpha$. Figure 20 shows the standard deviation. For the row corresponding to the proxy, the advantage corresponds to the adaptive proxy minus the vanilla proxy. For the rows corresponding to the reward vs expert, it corresponds to the reward of the adaptive policy minus the one of the vanilla policy. For the rows corresponding to the exploitability, it corresponds to the exploitability of the vanilla minus the one of the adaptive policy.

Figure 21 illustrates the convergence of Fictitious Play (FP) with common noise (Algorithm 1), the training loss of the expert learning to replicate the aggregated mean fields generated by the FP-outputted policies (Algorithm 2), and the training losses for the vanilla and adaptive parametric policies trained via Algorithm 3.

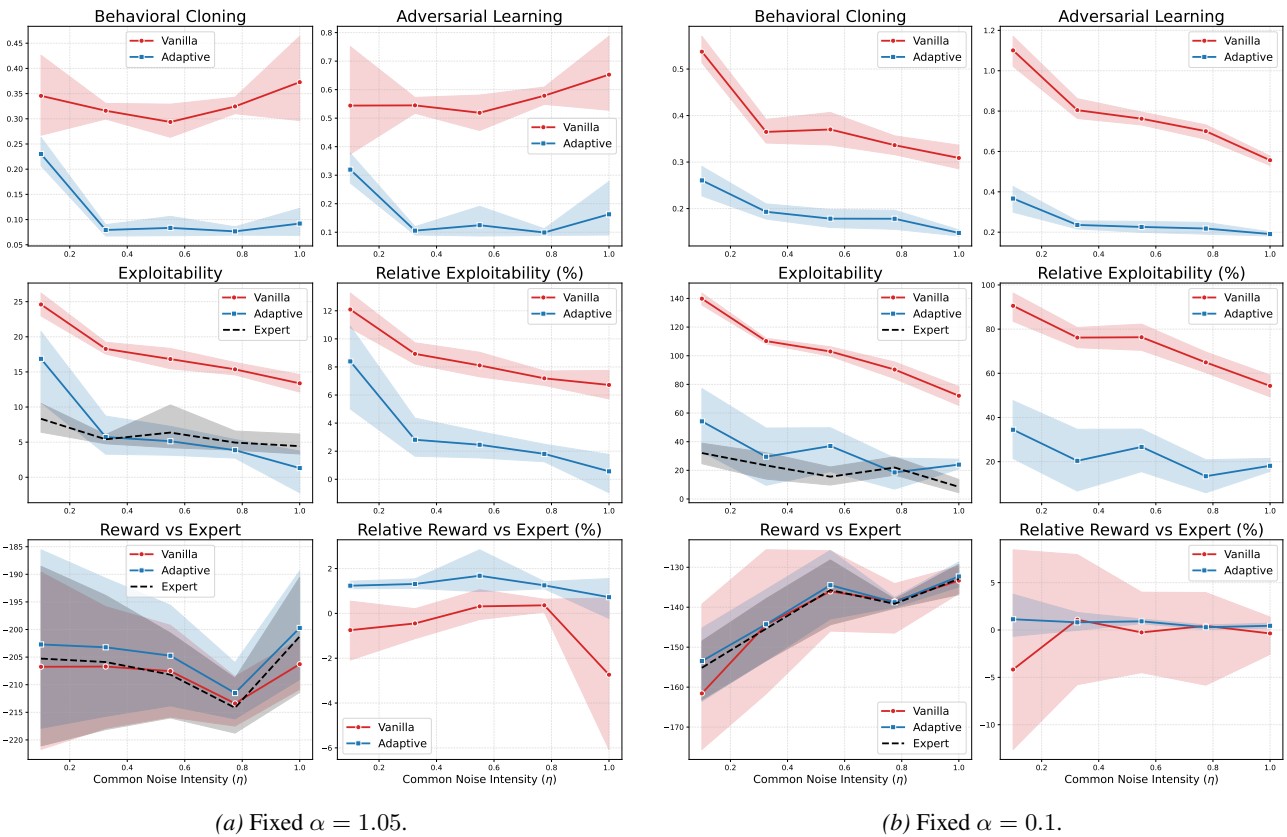

*(a)* Fixed $\alpha = 1.05$.               *(b)* Fixed $\alpha = 0.1$.

*Figure 18.* Performance metrics for 5 different runs in the Night Clubs example, with $\alpha$ fixed and $\eta$ varying.

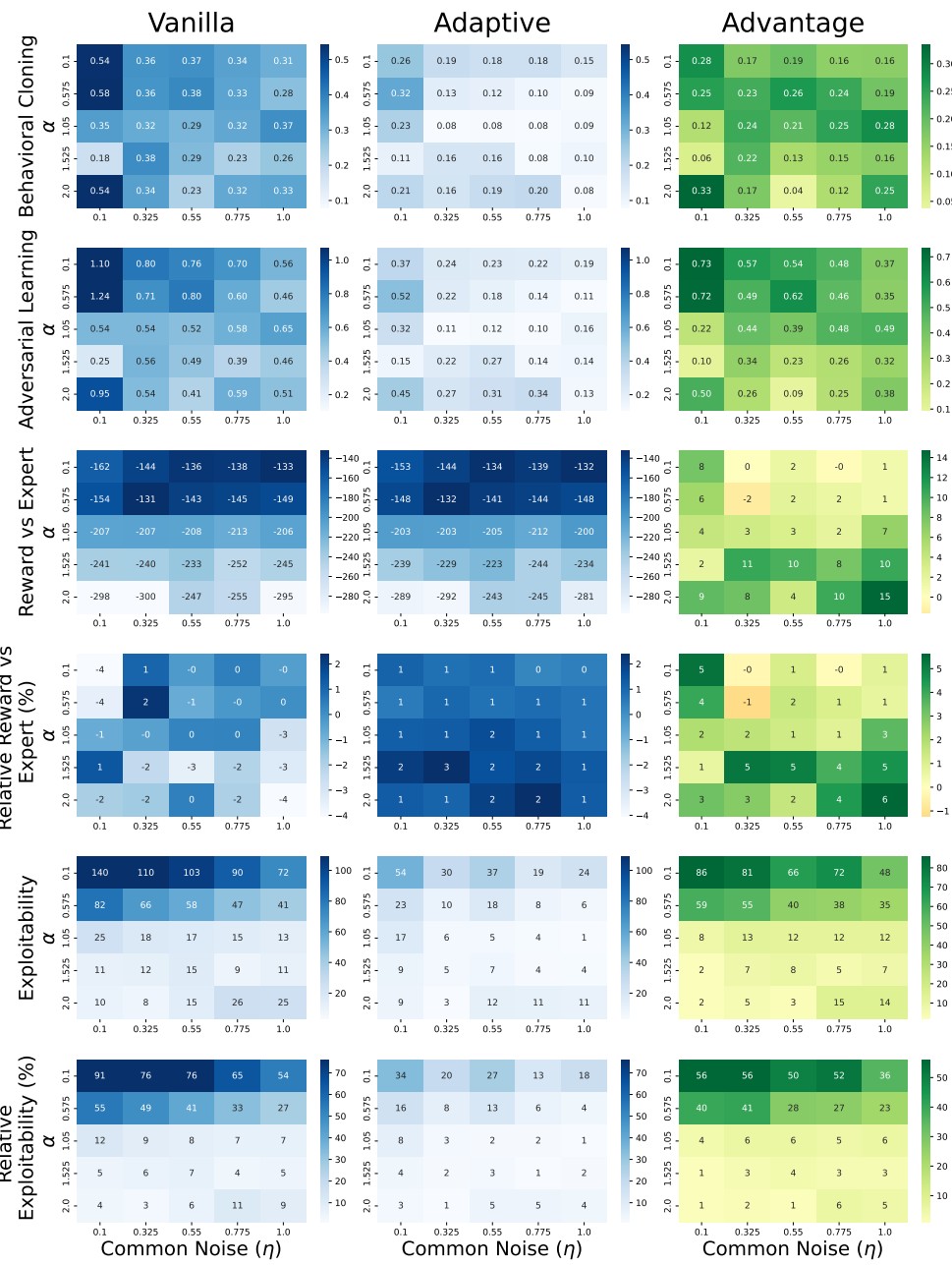

*Figure 19.* Night Clubs: Metrics Averages

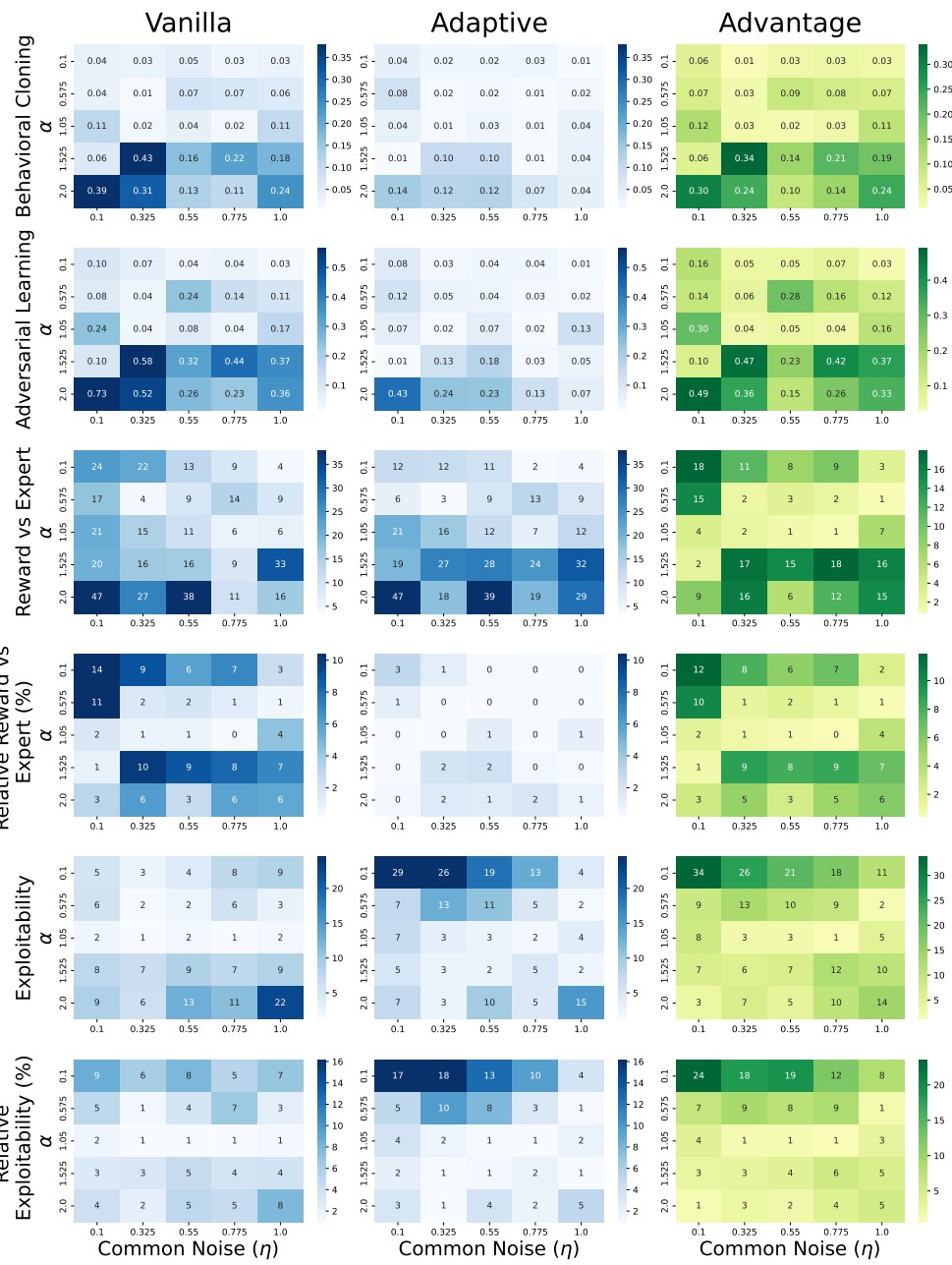

*Figure 20.* Night Clubs: Metrics STD.

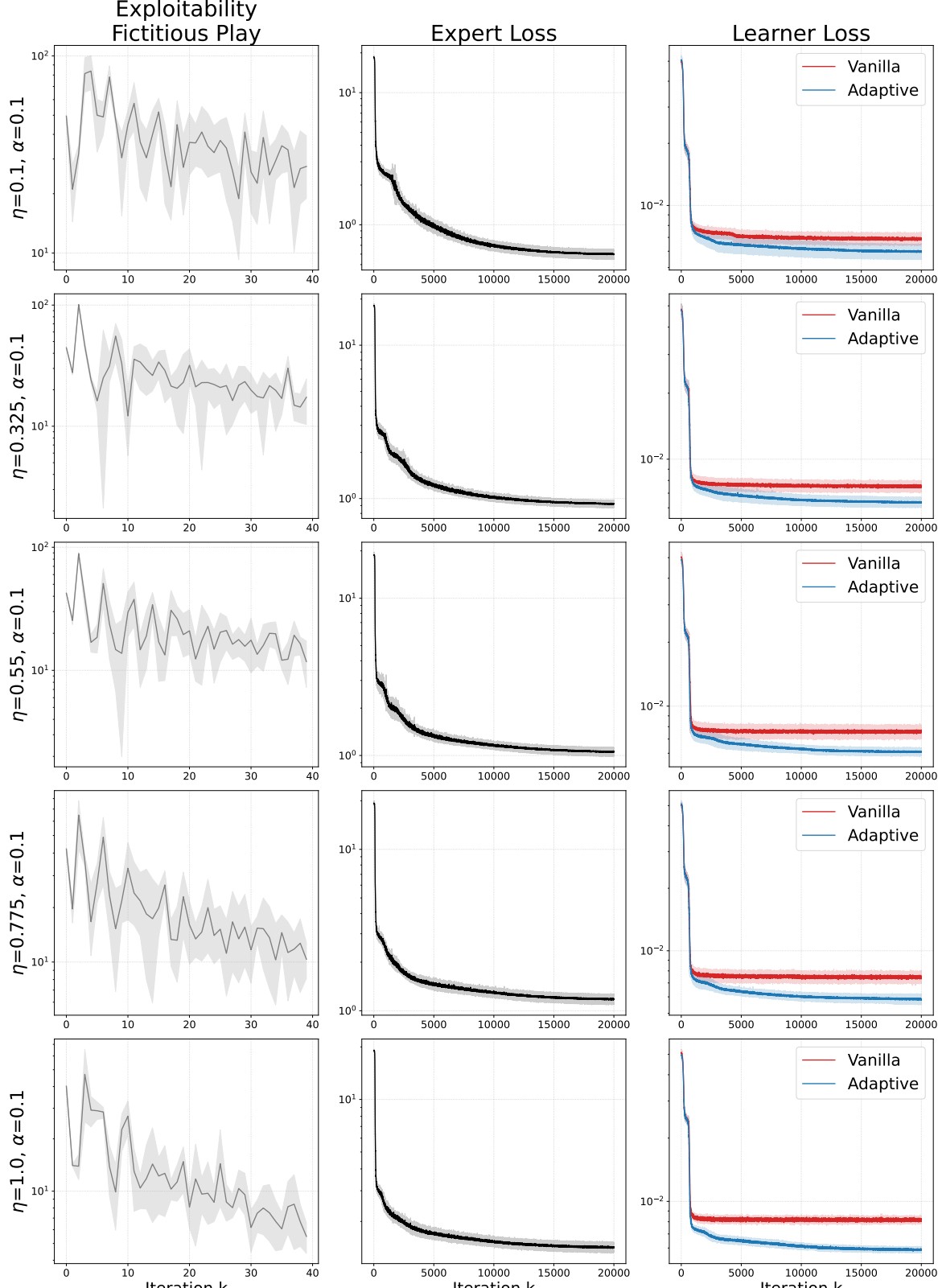

*Figure 21.* Night Clubs: Fictitious Play Exploitability Convergence, Expert Losses, Vanilla and Adaptive Losses ($\alpha = 0.1$).

