# OpenReview forum: "Population-Aware Imitation Learning in Mean-field Games with Common Noise"
_ICML.cc/2026/Conference — ICML 2026 regular_

### Official Review · Reviewer_tvFP · 2026-02-17

**Soundness:** 2
**Presentation:** 3
**Significance:** 2
**Originality:** 2
**Overall Recommendation:** 4
**Confidence:** 3

**Summary:**

The paper studies imitation learning in mean field games under common noise. The authors propose two imitation metrics, behavior cloning and adversarial divergence, and connect them to exploitability and certain performance gap relative to the expert. They also introduce algorithms based on fictitious play that computes population-aware expert policies and validate the algorithm performance in a few environments.

**Compliance With Llm Reviewing Policy:**

Affirmed.

**Final Justification:**

My opinion of the paper was that it is right on the borderline, and the authors' response has moved my stance slightly in the positive direction.

**Key Questions For Authors:**

See weaknesses above

**Limitations:**

No concerns

**Strengths And Weaknesses:**

The paper studies an interesting problem and identifies the limitation in the existing imitation learning methods for accounting for population dependency. Overall the paper is well-written and presents the motivation and main contributions reasonably well.

On the negative side, I have the following concerns on the significance and novelty of both theoretical and empirical results.

1) It is not unclear to see what I should take away from Theorem 3.6 and 3.7. A Nash equilibrium in a MFG is a point where $\pi$ is optimal in an MDP defined under the mean field induced by $\pi$. This says nothing about whether $V(\pi,\pi)$ is higher or lower than $V(\pi',\pi')$ for some other $\pi'$, so I am confused why we should compare $V(\pi^E,\pi^E)$ and $V(\pi^A,\pi^A)$.

2) The authors claim that common noise significantly complicates the error analysis compared to the deterministic setting. Can the authors pinpoint where the complication exactly arises in the analysis and what innovation is made to handle it? I went through the proofs of the main lemmas in the appendix and get the sense that all proofs use quite straightforward arguments.

3) The proofs of theorems do not provide meaningful information. It is perfectly fine for proof (sketch) not to be presented in the main paper. However, if the authors would like to present a proof sketch, they should make an effort to make it readable and informative.

4) Given the possibly limited theoretical contributions, and given the authors' claim that the algorithm is designed to be compatible with deep learning, I would expect that the authors conduct larger-scale experiments that provide solid and comprehensive insight on the algorithm performance. The three environments considered are rather small in scale and complexity.

5) Minor: The authors should proofread the figure layout. In the present manuscript, negative vspace around 1, 5, and 7 leads to formatting issues and a somewhat unpolished appearance.

---

> ### Author Rebuttal · Authors · 2026-03-30
>
> We thank Reviewer tvFP for the detailed evaluation and constructive feedback. We have addressed the specific concerns as follows:
>
> **1. Weakness 1 (Theorems 3.6 and 3.7: Performance Matching Bounds)**:  We thank the reviewer for this remark, which identifies a typographical error in the manuscript. The term $V(\pi^A, \pi^A)$ should indeed be $V(\pi^A, \pi^E)$. By evaluating the learner $\pi^A$ against the expert-induced mean field $\pi^E$, the theorems now correctly reflect the standard performance matching objective. We will update the manuscript accordingly.
>
> **2. W2 (Common Noise and Error Analysis)**: We appreciate the opportunity to clarify our perspective on the analytical challenges introduced by common noise. If our phrasing suggested that common noise "significantly complicates the error analysis," we would like to clarify our specific intended meaning. The integration of common noise required us to address two fundamental structural challenges:
> * **Policy Definition (mean-field dependency):** In contrast to previous works without common noise (e.g., Ramponi et al. (2023)), which consider policies as functions of the individual state only, common noise brings us to employ population-dependent policies. This has the advantage of letting the agents react to new distributions, but the fact that the space of probability distributions is continuous means that we need to pay attention to the regularity of the policy with respect to this continuous input. This makes the analysis more subtle than for purely discrete spaces (e.g., Ramponi et al. (2023)). We addressed this challenge by considering the class of policies with Lipschitz regularity.
> * **Proxy Definitions and Stochasticity:** There are multiple ways to extend imitation proxies to stochastic environments. We chose a definition that synchronizes the noise realizations in order to isolate the policy's response to the mean field itself. A proxy extension to stochastic settings, constitutes a novel contribution of this work.
>
> Our choice of policies and proxies is meaningful and our analysis is designed to fit these notions. Although deriving them was not straightforward, we are glad to know that the presentation is now "easy to follow."
>
>
> **3. W3 (Dimensionality and Scale of MFG Environments)**: We would like to stress that, although the individual player has only a few states and actions, in our context of MFGs with population-aware policies, the dimensionality corresponds to the one of the *population distribution*, which is an element of the simplex. So if the player has 20 states, the policy's input is of *dimension 20* (without taking into account the individual state). This is already well beyond what can be handled by tabular RL methods and justifies resorting to neural networks. We do not see any major bottleneck for scaling up our approach to larger problems (provided there is enough compute resources), but it is beyond the scope of this paper. We hope that the reviewer will agree that our examples are sufficient for an academic paper, with both theoretical and numerical contributions. Furthermore, we would like to stress that our models are much *larger than the ones considered* in other works on IL/IRL for MFGs, e.g. in Ramponi et al. (2023) (2 states), Chen et al. (2022) (10 states), or Anahtarci et al. (2025) (10 states).
>
> **4. W4 (Minor comments)**: Thank you for pointing these outs. Regarding the figure layout and formatting: We will remove the aggressive spacing and correct the layout of Figures 1, 5, and 7 to ensure a clean and professional presentation.
>
> We hope that these elements will answer the questions, in which case we would be grateful if you could consider raising your score. Otherwise, we will be happy to answer any follow-up questions.

---

> > ### Author Rebuttal · Reviewer_tvFP · 2026-04-01
> >
> > My opinion of the paper was that it is right on the borderline, and the authors' response has moved my stance slightly in the positive direction. I would like to adjust my rating accordingly.

---

### Official Review · Reviewer_bU21 · 2026-02-23

**Soundness:** 2
**Presentation:** 2
**Significance:** 2
**Originality:** 2
**Overall Recommendation:** 4
**Confidence:** 3

**Summary:**

This paper focuses on mean field games with common noise where noise influences the mean field distribution. Thus, in contrast to standard mean field games, the mean field remains stochastic in the limit which poses an additional modeling challenge. The authors consider the task of inferring an (unknown) reward function from the observation of an expert population. By employing imitation learning and considering two imitation metrics, the authors design a learning algorithm based on the well-established Fictitious Play algorithm. They conclude their paper by evaluating their approach on three different problem instances.

**Compliance With Llm Reviewing Policy:**

Affirmed.

**Final Justification:**

The authors have adequately addressed many of my concerns. Consequently, I have increased my score.

**Key Questions For Authors:**

My main questions are closely related to the weaknesses listed above:
1. What are the significant novel theoretical contributions of this paper? What are substantial differences to Ramponi et al. (2023)?
2. What are the main algorithmic contributions compared to existing Fictitious Play and Imitation Learning approaches?
3. As explained above, I do not understand why the paper spends so much time/space on the moderately informative experiments section. Did I miss important insights? What is the benefit of discussing three different experiments in detail if their results are very close to each other?

**Limitations:**

yes

**Strengths And Weaknesses:**

Strengths:
- The considered topic appears important, especially as the MFG literature on including empirical data is still limited
- Both the theoretical and analysis appear to be rather extensive

Weaknesses:
1. The theory section and its results seem to be very closely related to Ramponi et al. (2023). Consequently, the novelty of the mathematical contributions made here is limited.
2. Something similar goes for the algorithms: The numerical contributions appear to be heavily based on Fictitious Play and existing Imitation Learning algorithms for MFGs. While there is nothing wrong with that in principle, I find it hard to understand what major difficulties had to be overcome and what novel ideas were necessary to overcome them in the present work.
3. The authors use the last three pages for their empirical evaluation. Despite this extensive evaluation, the insights often remain rather obvious and/or repetitive: i) it is quite trivial that the vanilla policy (which does not take the MF into account) will not be able to react to a changing MF, ii) In general, all three experiments seem to yield very similar results: thus, I do not see why each experiment and its results need to be discussed in so much detail in the main part of the paper.
4. Adding to the repetitive nature of the experiments section, the authors "reuse" complete sentences multiple times, for example "To ensure statistical stability, for each parameter configuration $(\alpha, \eta)$, we perform 5 independent runs of the full experimental pipeline" is used in lines 322-324, 369-372, and 429-431. This brings up the question if there isn't any more important information to convey in the main part of the paper.

Minor comments:
1. There are several typos in the manuscript, for example: $\pi$ instead of $\pi^A$ (Definition 2.7, second line); "redirect" instead of "redirects (line 399); "are" instead of "is" (line 400)
2. The assumption of Lipschitz policies on page 3 should be formulated as an explicit assumption like Assumption 2.3 and 2.4. In its current form, this assumption can be easily overlooked.
3. Section 4.6: Why is a beta distribution chosen for the common noise?
4. Why is a "threshold effect" introduced in the Beach Bar and the Night Club environment?
5. The last line of the caption of Figure 5 collides with the main text. That should be fixed

---

> ### Author Rebuttal · Authors · 2026-03-30
>
> We appreciate the extensive review and the constructive remarks regarding our work. We hope that the following clarifications provide satisfying answers to your questions and further demonstrate the novel contributions of our paper. We want to stress that in real-world scenarios, common noise can affect populations but applying existing methods (with population-independent policies) in such situations fails to recover equilibria (as demonstrated in our experiments by the baseline). **Our work provides theoretical and algorithmic tools to tackle this challenge and perform imitation learning in the presence of common noise.**
>
> **1. W1 and Q1 (Theoretical Novelties):**
> First, the **common noise** complicates the analysis, leading us to consider population-aware policies and to define suitable proxies; due to space constraint, we refer to **Point 2** in our response to **Reviewer tvFP**. Furthermore, our paper addresses **performance matching**, which is important but not addressed by Ramponi et al. (2023).
>
> **2. W2 and Q2 (Algorithmic Contributions)**: There are actually two main contributions:
> * **IL Algo.**: Ramponi et al. (2023) does not provide any concrete learning algorithms; moreover their experiments are very-small scale (2 states). In contrast, we propose a practical learning algorithm and demonstrate its effectiveness across 3 environments, with carefully designed common noise.
>
> * **Policy Distillation**: A key innovation of our approach lies in the way we learn the equilibrium policy through distillation during Fictitious Play iterations (see Alg. 3). Returning a single policy is much more efficient than returning a list of policies as, e.g., in Perrin et al. (2020). This is crucial due to the stochasticity of the mean-field transitions, but can also be helpful in the absence of common noise.
>
> **3. W3 (Experiments)**: We reply to the 2 points:
>
> **i)** The reason we included this population-independent baseline is to show that our learned policy *does* perform better, which is a legitimate question; indeed, we tested simpler environments and found that in many cases, the equilibrium policy has very little sensitivity to the population, which is hard to guess by looking at the model; we carefully designed 3 environments in which the equilibrium policy depends heavily on the population, as evidenced by the gap with the baseline.
>
> **ii)** We would like to clarify the differences between our 3 environments. The motivation is to demonstrate that our approach is efficient in a variety of structures of common noise (CN):
> * **Exp. 1 (Concentration-Dependent)**: The CN is correlated with the population density (e.g., if the population is concentrated at state $0$, agents are more likely to be affected).
> * **Exp. 2 (Local Effects)**: The noise affects agent positions in a heterogeneous manner, leading to more chaotic local shifts.
> * **Exp. 3 (Block-Shift)**: The CN shifts specific clusters or "blocks" of agents while leaving others unaffected, representing a segmented environmental shock.
>
> In real-world applications, CN is complex and difficult to model explicitly. By showing consistent results across these three distinct scenarios, we validate that our population-aware IL method remains superior regardless of the specific CN manifestation.
>
> **4. W4 (Repetitions)**: We repeated these sentences for the sake of clarity but we will avoid repetitions in the final version.
>
> **5. Minor Comments (MC)**: Thank you for the constructive feedback. We will address the comments as follows:
>
> * **MC1: Typos:** All identified typos will be corrected.
>
> * **MC2: Assumption:** We will put this in a separate Assumption.
>
> * **MC3: Beta Distribution for CN:** We chose $Beta(\alpha, \beta)$ for its flexibility in modeling a wide range of perturbation structures:
>     * **Bimodal/Edge Cases ($\alpha < 1$):** By setting $\alpha = \beta < 1$, the mass concentrates at the boundaries ($0$ and $1$). This represents extreme environmental shocks, where the common noise frequently takes "all-or-nothing" values.
>     * **Unimodal/Stable Cases ($\alpha = 1$):** When $\alpha = \beta = 1$, the distribution is Uniform. If we move toward $\alpha, \beta > 1$, the mass concentrates around $0.5$. In our first environment, noise values near $0.5$ result in minimal system perturbation.
> * **MC4: Threshold Effects in Beach Bar/Night Club:** The "threshold effect" was introduced to model congestion and capacity constraints. In these environments, the utility of a state changes abruptly (e.g., a bar becomes less desirable once it reaches a certain density), and we wanted to demonstrate that our population-aware policies could accurately anticipate and respond to these non-linear dynamics.
> * **MC5: Spacing**: We will adjust the vertical spacing around the figures.
>
> We hope we answered your questions and we would be grateful if you considered raising your score. Otherwise, we will be happy to answer any follow-up questions.

---

> > ### Author Rebuttal · Reviewer_bU21 · 2026-04-02
> >
> > I thank the authors for their detailed and informative responses. Taking these answers and the opinions of the other reviewers into consideration, I will raise my score accordingly.

---

### Official Review · Reviewer_K2p3 · 2026-03-13

**Soundness:** 4
**Presentation:** 4
**Significance:** 2
**Originality:** 2
**Overall Recommendation:** 5
**Confidence:** 2

**Summary:**

This paper studies population-aware imitation learning in mean-field games with common noise. Treating common noise makes the problem harder, is an interesting addition and addresses a real gap in the literature. The paper adopts a discrete-time, finite-state space setting with Markovian policies. The main idea is that policies which are not population-aware can perform suboptimally under common noise. The main technical contributions regard BC and ADV bounds and are strong and rigorous. The paper also confirms the intuition that matching rewards is easier than recovering equilibrium: 'performance matching is robust but equilibrium recovery is fragile'.

**Compliance With Llm Reviewing Policy:**

Affirmed.

**Final Justification:**

I think the paper is a clear accept, with the author rebuttal further increasing the the confidence in my assessment.

**Key Questions For Authors:**

1. I can't tell if the current proof technique can be extended beyond finite state spaces and finite action spaces. Is there an obvious way to proceed, or the measure theory complications are significant?
2. Can the results be extended to the partial-observation setup?

**Limitations:**

The experiments are somewhat limited to synthetic, relatively small settings, without an extensive comparison to existing equilibrium solvers. The methodology is good, but unlikely to make a big impact.

**Strengths And Weaknesses:**

Introduction is well written and relevant works are cited. Although the proofs are quite involved and the technical results may be difficult to check fully, the ideas are outlined clearly and are easy to follow. The exponential versus polynomial bounds in Theorems 3.1 and 3.2 are illustrative. Both proofs and empirical results support the claims made in the paper .I appreciate that multiple runs are reported in the experiments section.

---

> ### Author Rebuttal · Authors · 2026-03-30
>
> We thank Reviewer K2p3 for the positive feedback regarding the writing quality and the rigor of our numerical experiments. We appreciate the opportunity to address your questions and provide further clarity on the potential extensions of our work. We hope that these clarifications provide the necessary intuition to bolster your confidence in the assessment of our work.
>
> **1. Q1 (Extension Beyond Finite Spaces)**: Our *theoretical* results appear to be extendable to continuous state-action settings. To formalize this extension, one would replace the $L_1$-norm with the Wasserstein distance on the probability space. For bounded state spaces, this metric induces a topology that ensures the space of probability measures is compact. However, we acknowledge that *numerical* implementation in the continuous setting presents a significant challenge. In this case, the probability measure becomes an element of an infinite-dimensional space rather than a finite-dimensional vector. Implementing an analog of our algorithm would require a suitable approximation of the population distribution.
>
> **2. Q2 (Extension to Partial-Observation Setups)**: This also an interesting future direction. In cases where "partial observation" means that the equilibrium policy is a function not of the whole population distribution but only of a quantity derived from it (e.g. moments or convolution with a kernel relative to the agent's position), the results seem to hold under the same assumptions. In fact, in the present imitation learning context, such a partial observation setting can be viewed as a restriction of the full-observation case. Indeed, learning a partial observation policy is less demanding, since the observation space is smaller. However, we must remain cautious, as the general existence of equilibria in MFGs under partial observations has not been extensively studied in the literature. One could also wonder how to perform imitation learning with partial observation on the data (e.g., trajectories subject to observation noise). Our algorithm is directly applicable in this setting, but one would need to carry out an analysis of the error propagation (how the noise affecting the data propagates in the learned policy and in the final exploitability). We believe this is doable and interesting in its own right but beyond the scope of the present paper.
>
> **3. Limitation (Dimensionality and Scale of MFG Environments)**: We would like to stress that, although the agent's state space is relatively small (e.g., 20 states), the real dimensionality of the problem is the mean field (a vector in dimension 20), because our policies take the distribution as an input. The fact that the mean field has a stochastic evolution in high dimension makes our environments intractable for tabular RL methods and justifies demonstrates the necessity of our deep learning approach. Last, we would like to stress that prior work on Imitation Learning for MFGs has very small experiments: for instance the experiment in Ramponi et al. (2023) had only 2 states while Chen et al. (2022) and Anahtarci et al. (2025) had only 10 states (and no common noise).

---

> > ### Author Rebuttal · Reviewer_K2p3 · 2026-04-01
> >
> > thanks, score upheld. good luck!

---

### Official Review · Reviewer_oTG4 · 2026-03-13

**Soundness:** 2
**Presentation:** 3
**Significance:** 3
**Originality:** 3
**Overall Recommendation:** 4
**Confidence:** 4

**Summary:**

The paper investigates MFG with common noise. In contrast to the standard MFG formulations where the independent noise leads to a deterministically propagated mean field, in the common noise case, all agents are affected at the same time, resulting to a mean field that becomes a stochastic process adapted to the common noise filtration. The problem being investigated in the following: a learner observes a population of experts playing a Nash equilibrium but does not know the reward function driving their behavior. The goal is to learn a policy that imitates the expert, ideally recovering a Nash equilibrium and matching the expert’s performance. Hence, two distinct learning objectives are pursued: first, recover a Nash equilibrium, and second, maximize the performance against an expert population. A numerical framework that uses generalized Fictitious Play and Deep Learning is proposed to compute expert population-aware policies. The results of the paper show that learning population-aware policies is crucial to avoid being misled by the randomness inherent in common noise.

**Compliance With Llm Reviewing Policy:**

Affirmed.

**Final Justification:**

The revision has helped me better understand the paper's contributions and limitations. My original score reflects this, so it has remained unchanged.

**Key Questions For Authors:**

1. Why is the 1-norm used in the divergence to compare the policies (in both the BC and ADV cases)? Will it make a difference if one used, say a KL-divergence?

2. What are the labels of the vertical axes in Figure 1?

3. (Q2) tries to answer the question of how far is πA from being a best response to πE. Theorems 3.5 and 3.6 try to answer this question by providing the value of the corresponding policies instead of the policies themselves. It seems to me that the authors answer a different question that the one posed as Q2. If the results in Theorems 3.5 and 3.6 that aim to provide bounds on performance are what one cares about in practice (which I believe is the case), then perhaps Q2 can be rephrased to better reflect this fact?

**Limitations:**

The main limitation is the a priori regularity assumptions needed for the resulting policies. The authors should comment how this can be ensured in practice. I believe there are several algorithms that can enforce smoothness constraints on the policies during NN training, although I am not sure if these can be utilized “out of the box” for this problem or whether a new, bespoke algorithm needs to be devised to make the approach more easily applicable in practice.

**Strengths And Weaknesses:**

Strengths

1. The paper addresses an interesting and relevant problem in MF literature.

2. The paper provides a theoretical and algorithmic framework for imitation learning in MFGs with common noise, bridging a significant gap in the current state of knowledge.

3. Good choice of numerical examples that demonstrate the theory.

Weaknesses

1. The results depend strongly on certain regularity assumptions regarding the agent policies wrt to the mean field. Since the policy is yet to be designed, this Lipschitz regularity needs to be enforced explicitly.

---

> ### Author Rebuttal · Authors · 2026-03-30
>
> We thank Reviewer oTG4 for the detailed review and the insightful technical questions. We appreciate the recognition that our work bridges a significant gap in the literature and provides a strong theoretical and algorithmic framework. We hope the following response provides satisfying answers to your questions, specifically regarding the limitations raised.
>
>
> **1. Weakness (Policy Regularity):** Thank you for highlighting this important point. We will isolate the regularity assumption in a separate Assumption. We would like to stress that:
> * **(i)** Several **prior works** assume Lipschitz-continuous policies, such as (Asadi et al., 2018) and (Barbara et al., 2025) in the context of RL  (these references are at the end of our response), and (Cui et al., 2024) in the context of mean field control problems (see their Assumption 2 and the references therein), which is closely related to ours. Furthermore, there exists methods to ensure this property for neural networks, see e.g. Gouk et al. (2021).
> * **(ii)** We believe it is possible to prove that the **equilibrium policy is indeed Lipschitz**, under suitable regularity conditions on the dynamics and the reward function. In continuous-time MFGs, this Lipschitz property is a consequence of the regularity of the Master equation solution, which is obtained by arguments based on partial differential equations (PDEs). For example, Theorem 2.8 in (Cardaliaguet et al., 2019) shows that the Master equation solution $U$ is $\mathcal{C}^1$. In the discrete-time setting, PDE-arguments cannot be used. In the context of discrete-time mean-field cooperative problems (also called mean field control), such results have been established. For instance Proposition 4.2 in (Motte and Pham, 2022) show that the value function is Lipschitz with respect to the distribution. We expect similar arguments to be useful in the context of MFGs.
>
>
> **2. Question 1 (Choice of the Norm, $L_1$ vs. KL):** Via Pinsker's Inequality, the $L_1$-norm is bounded by the KL-divergence. Consequently, if the Lipschitz assumptions were reformulated with respect to the KL-divergence, our theoretical results would still hold.
>
> **3. Q2 (Figure 1 Axis Labels)**: We apologize for the omission. The vertical axes in Figure 1 represent, respectively: the *BC Proxy Value*, the *ADV Proxy Value*, the *Relative Exploitability Value*, and the *Relative Reward vs. Expert Value*. We will update the figure in the revised manuscript.
>
> **4. Q3 (Theorems 3.5/3.6):** We are very grateful for this observation. There was indeed a typographical error in the statement of Theorems 3.5 and 3.6. The term $|V(\pi^A, \pi^A)|$ should indeed be $|V(\pi^A, \pi^E)|$. This correction aligns the theorems with the original intent of our Research Question 2, which is to measure how closely the agent's performance matches the expert's behavior when interacting with the expert population. We will correct this in the final version of the paper.
>
> **References:**
>
> * Asadi, K., Misra, D., \& Littman, M. (2018). Lipschitz continuity in model-based reinforcement learning. In International conference on machine learning (pp. 264-273). PMLR.
>
> * Gouk, H., Frank, E., Pfahringer, B., \& Cree, M. J. (2021). Regularisation of neural networks by enforcing lipschitz continuity. Machine Learning, 110(2), 393-416.
>
> * Barbara, N. H., Wang, R., \& Manchester, I. R. (2024, September). On robust reinforcement learning with Lipschitz-bounded policy networks. In Symposium on Systems Theory in Data and Optimization (pp. 137-152). Cham: Springer Nature Switzerland.
>
> * Cardaliaguet, P., Delarue, F., Lasry, J. M., \& Lions, P. L. (2019). The master equation and the convergence problem in mean field games. Princeton University Press.
>
> * Cui, K., Fabian, C., Tahir, A., \& Koeppl, H. (2024) Major-Minor Mean Field Multi-Agent Reinforcement Learning. In Forty-first International Conference on Machine Learning.
>
> * Motte, M., \& Pham, H. (2022). Mean-field Markov decision processes with common noise and open-loop controls. The Annals of Applied Probability, 32(2), 1421-1458.

---

> > ### Author Rebuttal · Reviewer_oTG4 · 2026-04-03
> >
> > I would like to thank the authors for their efforts in revising the paper.

---

### Decision · Program_Chairs · 2026-04-30

**Decision:**

Accept (regular)

**Comment:**

After the rebuttal, all reviewers concur in recommending weak acceptance